# A giant virus infecting the amoeboflagellate *Naegleria*

Patrick Arthofer [1,2], Florian Panhölzl[1], Vincent Delafont [3], Alban Hay [3], Siegfried Reipert[4], Norbert Cyran[4], Stefanie Wienkoop[5], Anouk Willemsen [1], Ines Sifaoui[6,7], Iñigo Arberas-Jiménez[6], Frederik Schulz [8], Jacob Lorenzo-Morales[6,7] & Matthias Horn [1] ✉

Giant viruses (*Nucleocytoviricota*) are significant lethality agents of various eukaryotic hosts. Although metagenomics indicates their ubiquitous distribution, available giant virus isolates are restricted to a very small number of protist and algal hosts. Here we report on the first viral isolate that replicates in the amoeboflagellate *Naegleria*. This genus comprises the notorious human pathogen *Naegleria fowleri*, the causative agent of the rare but fatal primary amoebic meningoencephalitis. We have elucidated the structure and infection cycle of this giant virus, *Catovirus naegleriensis* (a.k.a. Naegleriavirus, NiV), and show its unique adaptations to its *Naegleria* host using fluorescence in situ hybridization, electron microscopy, genomics, and proteomics. Naegleriavirus is only the fourth isolate of the highly diverse subfamily *Klosneuvirinae*, and like its relatives the NiV genome contains a large number of translation genes, but lacks transfer RNAs (tRNAs). NiV has acquired genes from its *Naegleria* host, which code for heat shock proteins and apoptosis inhibiting factors, presumably for host interactions. Notably, NiV infection was lethal to all *Naegleria* species tested, including the human pathogen *N. fowleri*. This study expands our experimental framework for investigating giant viruses and may help to better understand the basic biology of the human pathogen *N. fowleri*.

Viruses are the most common biological entities on our planet, infecting virtually all cellular organisms[1]. While viruses were originally believed to be smaller than 200 nm in size, the discovery of the first giant virus, *Acanthamoeba polyphaga mimivirus*, with a genome and particle size comparable to bacteria has changed our perception of the viral world and the complexity of viral particles and their genomes[2].

All known giant viruses belong to the *Nucleocytoviricota* phylum (formerly known as Nucleocytoplasmic large DNA viruses, NCLDV) and have double-stranded DNA genomes. Environmental genomics revealed a huge, previously unseen diversity of giant viruses, and recovered metagenome assembled genomes (MAGs) with a nearly complete set of translation genes[3,4] and unexpected metabolic potential[5] from virtually every conceivable environment globally.

[1]University of Vienna, Centre for Microbiology and Environmental Systems Science, Division of Microbial Ecology, Vienna, Austria. [2]University of Vienna, Doctoral School in Microbiology and Environmental Science, Vienna, Austria. [3]Ecologie et Biologie des Interactions Laboratory (EBI), Microorganisms, hosts & environments team, Université de Poitiers, UMR CNRS, Poitiers, France. [4]University of Vienna, Research Support Facilities UBB, Vienna, Austria. [5]University of Vienna, Department of Functional and Evolutionary Ecology, Division of Molecular Systems Biology, Vienna, Austria. [6]Instituto Universitario de Enfermedades Tropicales y Salud Pública de Canarias, and Departamento de Obstetricia y Ginecología, Pediatría, Medicina Preventiva y Salud Pública, Toxicología, Medicina Legal y Forense y Parasitología, Universidad de La Laguna, Tenerife, Islas Canarias, Spain. [7]Centro de Investigación Biomédica en Red de Enfermedades Infecciosas (CIBERINFEC), Instituto de Salud Carlos III, Madrid, Spain. [8]DOE Joint Genome Institute, Lawrence Berkeley National Laboratory, Berkeley, USA. ✉e-mail: matthias.horn@univie.ac.at

Metagenomics has thus been fundamentally important for our understanding of giant virus diversity, biogeography, and evolution[3,5–8]. Yet, incomplete genome sequences, the lack of direct information about the host organism, and the absence of experimentally accessible isolates represent inherent limitations of this approach.

Virus biology can ultimately be only understood in the context of the host organism, but the very low number of giant virus isolates available so far are limited to only a few ameba, flagellate, and algae host genera[2–4,9–14]. Despite that, the known giant virus isolates differ significantly in the morphology of their capsid, from icosahedral to ovoid structures deviating from the classical viral capsid structure. These morphological features affect all parts of the viral infection cycle, from adhesion, entry, and DNA replication, to particle assembly and egress. Infections with giant viruses and the lysis of their hosts can have ecosystem scale consequences, for instance through the termination of algal blooms[15].

Microbial eukaryotes are responsible for causing some serious diseases in humans and livestock[16,17]. One such disease is primary amebic meningoencephalitis (PAM), which is caused by the amoeboflagellate *Naegleria fowleri*. PAM is a rare but incurable disease that occurs globally and has a case fatality rate of more than 98%[16]. *Naegleria* species thrive especially well in warm water bodies, even at geothermal aquatic sites[16]. Though it was speculated that giant viruses share an evolutionary history with members of the genus *Naegleria*[18], no giant virus infecting these protists has been found yet.

This study reports on the discovery of a giant virus isolate infecting *Naegleria* species, including the human pathogen *N. fowleri*. We investigated the morphology, genome, proteome, and host range of this novel giant virus. Naegleriavirus is currently the fourth Klosneuvirus isolate and shows unique adaptations to its *Naegleria* host.

## Results

A highly lytic virus was isolated from a nitrifying reactor of a wastewater treatment plant in Klosterneuburg, Austria, by co-cultivation with *Naegleria clarki*. The first Klosneuvirus MAG originated from this reactor[3]. The virus isolate was subsequently maintained using *N. clarki* under monoxenic conditions with *Escherichia coli*.

### The Naegleriavirus genome and proteome

The complete, linear double-stranded DNA genome of the *Naegleria clarki* infecting virus measures 1,163,307 bp and contains 1000 predicted coding sequences (CDS), 580 on one and 520 CDS on the other strand, respectively (Fig. 1a, Supplementary data set 1).

Phylogenetic analysis of three *Nucleocytoviricota* core genes revealed that the virus belongs to the family *Mimiviridae*, subfamily *Klosneuvirinae*, and represents a sister clade of a MAG, which was recovered from a wastewater bioreactor in Cape Town, South Africa and named Catovirus CTV1 (Fig. 1b, Supplementary data set 2)[3]. In reference to its ability to infect *N. clarki*, we, therefore, propose to classify the new viral isolate as *Catovirus naegleriensis*, with the trivial name Naegleriavirus and the acronym NiV. Most of its genes (68.3%)

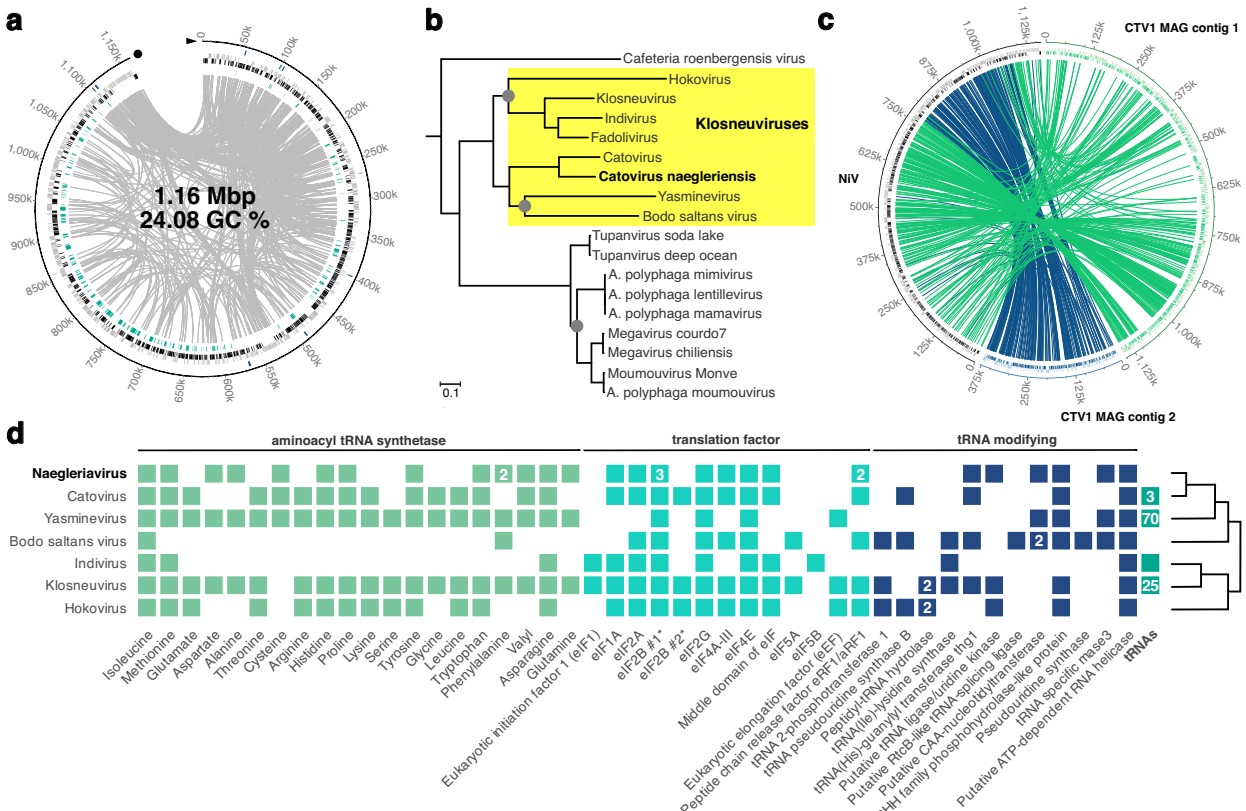

**Fig. 1 | The *Catovirus naegleriensis* (Naegleriavirus) genome, phylogeny, and translation modules. a** The linear double stranded DNA genome with 1000 predicted CDS. From outer to inner track: blue, putative host-derived genes; gray, black 480 and 520 CDS on the two strands, respectively; cyan, proteins detected in the virion; gray links represent homologous CDS. **b** Maximum likelihood tree constructed with three *Nucleocitoviricota* core genes, DNA polymerase family B, A18-like helicase, and poxvirus late transcription factor VLTF3. The tree is rooted with the Asfarviridae as an outgroup. The full version of the tree is available as Supplementary data set 2. **c** Circular representation of homologs shared between the Naegleriavirus (NiV) genome and the two contigs of the Catovirus CTV1 MAG; colored links depict bi-directional best BLAST hits. **d** Translation-related genes found in Naegleriavirus compared to other Klosneuviruses. Numbers in the boxes indicate the number of homologs. The relationship of the viruses is indicated as a schematic representation of the phylogenetic tree in panel **b**.

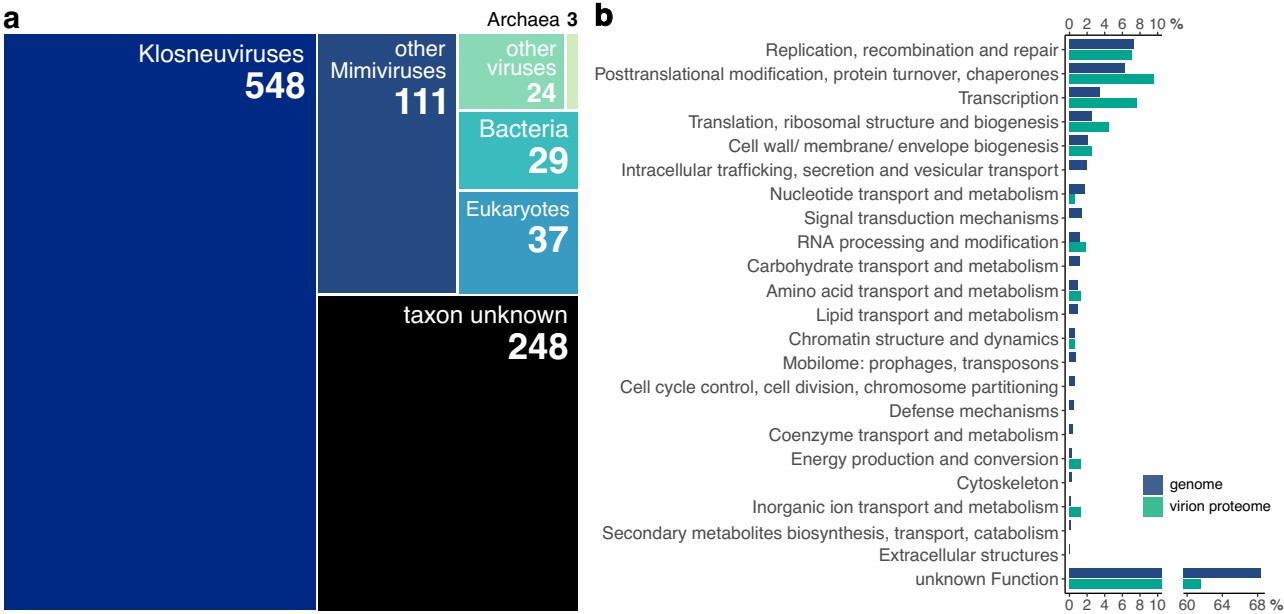

**Fig. 2 | The Naegleriavirus genome and its virion proteome. a** Taxonomic classification of 1000 predicted NiV genes based on similarity to viruses and cellular organisms using BLASTp against the NCBI nr database. **b** Functional categorization of NiV genes based on an automated PROKKA annotation, InterproScan, eggNOG, and manual curation. The percentages of genes/proteins in the genome/proteome for each COG category are depicted. Annotation details are available in Supplementary Data Sets 1, 3.

show the highest similarity with virus genes (Fig. 2a), while 47.7% have homologs (best bidirectional BLAST hit) in the genome of Catovirus CTV1 (Fig. 1c)[3]. In total, 24.8% are orphan genes with no match in sequence databases, 3.7% were assigned to Eukaryota, 2.9% to Bacteria, and 0.3% to Archaea (Fig. 2a).

A functional analysis of NiV genes that combined an automated PROKKA annotation with BLASTp, InterproScan, eggNOG searches, and manual curation left 683 CDSs (68.3%) with a general function prediction only or of unknown function (Fig. 2b, Supplementary data set 1). Those proteins for which a function was inferred were predicted to play a role primarily in replication, recombination and repair; transcription and translation; posttranslational modification and protein turnover, including 12 heat shock proteins; cell envelope biogenesis; and intracellular trafficking, secretion, and vesicle transport (Fig. 2b).

We next purified NiV virions from *N. clarki* cultures for proteomic profiling, which resulted in the identification of 159 NiV proteins. Out of those, 123 proteins were identified with at least two peptides, and 36 proteins were identified with one peptide hit only, but this one peptide was then detected in all five replicates. Similar to other giant virus virion proteomes, more than half of these proteins ($n = 96$) are of unknown function (Fig. 2b, Supplementary data set 3)[4,19,20]. Yet, all mimivirus virion key proteins were present in the NiV virion, including structural proteins such as the major capsid protein and the core protein, as well as the DNA-dependent RNA polymerase subunits and proteins involved in replication. The most important aspects of our genomic and proteomic analysis are summarized below.

The NiV genome encodes four major capsid domain-containing proteins (CDS 686, 687, 688, and 706) and one major core protein 4b (CDS 720). In addition, two genes were identified as minor capsid proteins using remote homology searches[21], which suggest they represent a tape measure protein (CDS 824) and a penton protein (CDS 654), respectively. In the virion proteome, only one major capsid domain-containing protein (CDS 706) was abundantly found, thus likely representing the major capsid protein (MCP) of NiV. Its homologs might represent low abundant proteins involved in specific capsid features[22].

A hallmark of Klosneuvirus genomes is the presence of translation-related genes[3]. The NiV genome is no exception to this trend and contains in total 31 translation-related genes (Fig. 1d). This includes 14 tRNA synthetases and a diversity of eukaryotic initiation factors. However, like in the Bodo saltans virus genome[11] no tRNAs but several tRNA repair genes were found (Fig. 1d). From this set of translation-related proteins, the tRNA(His) guanylyltransferase (CDS 856), the prolyl-tRNA synthetase (CD 312), and several putative eukaryotic translation initiation factors were detected in the virion proteome (Supplementary data set 3).

With respect to genes functioning in cellular metabolism, a BlastKOALA search and KEGG mapper reconstruction resulted in 176 hits in 184 KEGG pathways. NiV encodes an acyl-CoA synthetase AMP-binding enzyme for beta-oxidation (CDS 832), a function also found in the *Naegleria* (host) genome (Supplementary Tables 1, 2). The two genes needed for polyamine biosynthesis are present in the NiV genome. A glycan metabolism pathway involving capsular polysaccharide biosynthesis protein Cap F (CDS 264), putative dTDP-d-glucose 4,6-dehydratase (CDS 302), and UDP-N-acetylglucosamine 2-epimerase WecB-like protein (CDS 303) is almost complete, only missing one enzyme among the annotated genes, namely a putative aminotransferase.

During their replication cycle, giant viruses manipulate the vesicle trafficking of the host[23]. Towards this end, NiV encodes for two SNARE complex proteins (CDS 76, 229), which enable membrane fusion of transport vesicles and their targets. Furthermore, a vesicle fusing ATPase (CDS 962), dynamin proteins (CDS 14, 15, 373), and additional small GTPases (CDS 237, 396, 417, 494, 531) are present (Supplementary Table 1). The NiV SNARE proteins and the ATPase are most similar to *Naegleria* proteins. The closest homologs of the small GTPases are eukaryotic, Catovirus, or Tupanvirus genes. Dynamin-related proteins of giant viruses have been shown recently to be involved in the remodeling of mitochondrial membranes during infection[24]. Indeed, all three predicted NiV dynamins are similar to other predicted Nuclecytoviricota dynamin-like proteins. Yet, in our TEM analysis we did not observe any striking changes in mitochondria morphology during NiV infection. An additional set of proteins putatively acting in

the manipulation of host cellular pathways are repeat domain-containing proteins involved in protein-protein interactions. These are encoded primarily at both ends of the NiV genome, which are characterized by an increased amount of homologous regions (Fig. 1c, the inner gray links). This is especially pronounced at the 5' end of the genome with eight ankyrin repeat domain-containing proteins (CDS 979−986) and nine leucine-rich repeat domain-containing proteins (CDS 987−995). Consistent with the putative role of these proteins in host cellular pathway modulation, they were largely absent in the NiV virion proteome, except for the small GTPase (CDS 237)[24].

A peculiar feature of giant viruses is that they can be infected by so-called virophages[25]. Virophages depend on the viral factories of their giant virus hosts for replication and are thereby true viruses of viruses[25,26]. Some giant viruses defend themselves against virophages with MIMIVIRE, a giant virus defense system containing distinct repeat regions[27]. The NiV genome comprises two repeat regions that were not predicted as protein-coding, the first spanning over 351 nucleotides (278353−278704, between CDS 240 and 241) with six repetitive units, each with the sequence AATATAACAGATGATGGATTAAA separating different spacer sequences, and the second with a length of 593 nucleotides (1132715−1133308, between CDS 984 and 985) with eight repetitive motifs, each with the sequence TTTGAAATGGATGTTAG AAAATAATTT separating different spacer sequences. A homology search with both repeats and spacer sequences yielded a diverse range of low similarity hits to prokaryotes and eukaryotes. The first repeat gave a hit to a metagenome sequence related to a Catovirus CTV1 MAG. Notably, we observed no sequence similarity to known MIMIVIRE elements or sequences from known virophages. We also failed to detect protein-coding genes with similarity to known MIMIVIRE-associated genes[27] in the immediate neighborhood of the repeat regions.

Gene transfer and acquisition is a fundamental process in the evolution of giant virus genomes[28]. A number of agents of gene mobilization have been described[28], including transpovirons and virophages in giant viruses that infect Amoebozoa. While these were not detected in the NiV genome, we found seven transposases or transposase domain-containing proteins and seven intein-containing proteins, including DNA binding proteins involved in replication and transcription (Supplementary Table 1). A role of these types of mobile genetic elements in competition against other viruses has been proposed[11]. Contrary to Bodo saltans virus and other giant viruses[11], we did not detect any self-splicing introns in NiV.

The NiV genome encodes 37 proteins with high sequence similarities to protist genes (Supplementary Table 2). We performed phylogenetic analyses of all proteins for which a functional prediction was available. This revealed that 16 NiV proteins were most closely related to non-*Naegleria* eukaryotes, and 11 proteins were either part of a *Naegleria* clade or represented a sister branch to *Naegleria* proteins (Fig. 3, Supplementary files), suggesting that the respective genes were acquired from a *Naegleria* or other protist hosts. These putatively host-derived genes are associated with protein turnover and translation, replication, signal transduction, cell cycle, and apoptosis inhibition (Supplementary Table 2). Among those is a HSP70 chaperone (CDS 426) with homologs in different *Mimiviridae* (Fig. 3e), confirming an earlier hypothesis that giant viruses share an evolutionary history with members of the genus *Naegleria*[18]. In addition, the virus potentially acquired a mitochondrial chaperone (CDS 80) (Fig. 3c) from its host, and a second HSP70 protein (CDS 734), shared with the Catovirus CTV1 MAG, was presumably acquired from a non-*Naegleria* eukaryote. NiV likely acquired the vesicle fusing ATPase (CDS 962) and the two putative target SNARE proteins (CDS 76, CDS 229; Fig. 3d and Supplementary files) to interact with the intracellular vesicle system of the host. Furthermore, NiV acquired an apoptosis inhibition factor (Bax-1, CDS 074), and an ATPase-domain containing protein from the *Naegleria* host (CDS 038, Figure 3ab), both detected in the virion proteome. The apoptosis inhibition factor Bax-1 was only detected as one

peptide in the proteome (yet in all five replicates), which might be due to the membrane-associated location of this protein.

## Viral replication cycle and morphology

We next studied the replication cycle of NiV in *Naegleria clarki* with light microscopy (Supplementary Fig. 1), fluorescence in situ hybridization (FISH), and nucleic acid staining using 4',6-diamidino-2-phenylindole (DAPI) (Fig. 4). In the early phase of NiV infection, no differences to uninfected amoebae were observed with FISH (Fig. 4a). However, already 1 hour post-infection (hpi) (Fig. 4b), small DAPI-stained structures appeared in the cytoplasm of the ameba host cell. Subsequently, the viral DNA replication of the virus led to bright DAPI-stained viral factories, which sometimes outsized the nucleus (Fig. 4c). Finally, 8 hpi viral particles accumulated near the viral factory (Fig. 4d) and were released by host cell lysis.

To investigate the NiV particle and replication cycle in more detail and compare it to other mimiviruses, we performed transmission electron microscopy (TEM) of sectioned resin embeddings based on both chemical fixation as well as cryofixation and freeze substitution techniques (Figs. 5, 6 and Supplementary Fig. 2). The NiV virions have a diameter of ~500 nm and consist of electron-dense round viral cores previously described as containing protein-complexed DNA and proteins bounded by the core wall[29,30]. The core wall is surrounded by five distinct layers separated by less electron-dense interstices (Fig. 5a). Analogous to previous observations for Mimivirus particles[31] these represent the inner membrane, a proposed additional membrane, the inner capsid shell, and the electron-dense hexagon-shaped capsid shell covered by the fiber layer. A single stargate, a thickened structure representing the delivery portal, is located at one vertex of the capsid (Fig. 5b). Close to the stargate structure within the virion, the extra membrane sac[29,32,33] was observed (Fig. 6b).

NiV virions, similar to many other giant viruses infecting amoebae[34], appear to enter the host cell through phagocytosis. After uptake, the viral particles are located in the phagosome, sometimes surrounded by multilamellar structures (Fig. 6a, d). We observed closed and open stargates (Figs. 5b, 6b), so we assume that similar to other mimiviruses[31], the NiV stargate changes its conformation from closed to open (Fig. 5b). The opening of the stargate allows for the fusion of the virion's inner membrane with the phagosome membrane and results in the release of the viral core into the host cytoplasm (Fig. 6c), leaving behind an empty capsid (Fig. 5c). Cross sections through virions undergoing this process suggest that the extra membrane sac (EMS) is related to fusion of the inner membrane with the phagosome membrane, potentially involved in the creation of a channel for the release of parts of the EMS and the viral core (Fig. 6c). The protein content of the extra membrane sac is thought to be involved in the formation of the viral factory and potentially important for early stages of Mimivirus infection[30,35]. As shown for Acanthamoeba polyphaga mimivirus, the NiV viral core presumably gives rise to the cytoplasmic viral factory (Fig. 6e).

At later time points (8 hpi), the fully grown viral factory buds out virions on its periphery (Fig. 6e). Overall, infected *Naegleria* cells produce large amounts of extracellular laminar structures (Fig. 6f, Supplementary Fig. 2). We also observed various ameba cells without detectable viral factories with decaying nuclei, and encysting cells devoid of virions (Supplementary Fig. 2). Some ameba cells appeared to continue to phagocytose new virions during active virion production (Fig. 6f).

## Host range, viral temperature range, and infection dynamics

As *Naegleria* species show large differences in temperature preferences, ranging from 15 to 46 °C[36], we next investigated the NiV host range and temperature sensitivity. To reduce the additional layer of complexity in infection experiments, which are caused by the presence of *E. coli* as a food source for the ameba host, we initially tried to infect

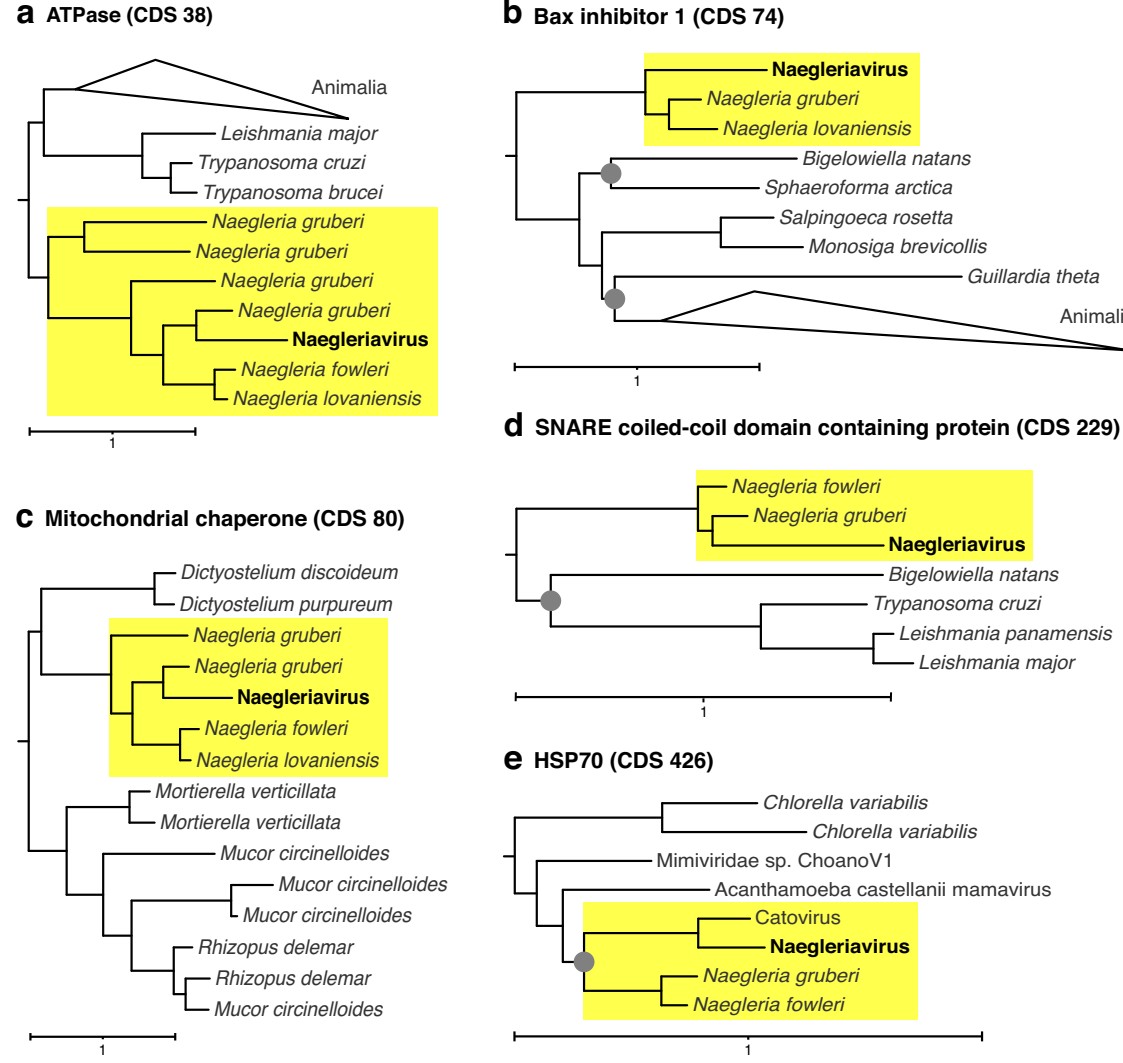

**Fig. 3 | Naegleriavirus genes potentially acquired from *Naegleria* hosts.**
Approximated maximum-likelihood phylogenetic trees are shown for genes
encoding an **a** ATPase-domain containing protein, **b** Bax 1 inhibition factor 1,
**c** mitochondrial chaperone, **d** SNARE coiled-coil domain-containing protein, and
**e** HSP70. Bootstrap values lower than 0.8 are depicted as gray circles. Only the
relevant subtrees, including Naegleriavirus genes are shown; full versions of the
trees are available as Supplementary Files.

*N. clarki* cultures in rich media devoid of bacteria. While amoebae grew
perfectly well under these axenic conditions, to our surprise, no viral
infection or replication was observed. We thus continued to work with
monoxenic ameba cultures and *E. coli* as a food source.

NiV induced infection phenotypes, such as cell rounding and lysis,
in all tested species of the genus *Naegleria*, including the human
pathogen *Naegleria fowleri*. NiV could not replicate in the typical
Amoebozoa hosts of other Mimiviruses, that are *Acanthamoeba castellanii* and *Vermamoeba vermiformis*. NiV also failed to replicate in
other heteroloboseans such as *Vahlkampfia* and *Tetramitus*, suggesting that the NiV host range is limited to members of the genus *Naegleria*. (Fig. 7). However, as we focused on host cytopathic effects in
our experiments, we cannot exclude low levels of virion production
without host lysis.

To study the influence of temperature on viral replication, *N.
clarki* and *N. lovaniensis*, adapted to ambient and elevated temperatures, respectively, were infected at 20, 30, and 40 °C. The virus
could successfully replicate in both ameba species at 20 and 30 °C
(Fig. 8a, b). The uninfected *N. clarki* population collapsed at 40 °C. *N.
clarki*'s inability to grow at 40 °C explains why the virus could not
replicate in this species at this temperature (Fig. 8c, Supplementary

Fig. 3). Counter to *N. clarki*, the thermophilic *N. lovaniensis* grew well at
40 °C in the absence of the virus. In the presence of NiV, *N. lovaniensis*
cell numbers remained notably stable during the entire infection
experiment (fold change -1 in Fig. 8d, Supplementary Fig. 3). This
shows that the virus impaired *N. lovaniensis* growth at 40 °C but did
not cause a decline of the ameba population. Even though NiV suppressed the growth of *N. lovaniensis*, this did not coincide with an
increase in virus abundance. On the contrary, the NiV genome copy
numbers fell below the starting numbers (Fig. 8b). This indicates that
viral particles entered *N. lovaniensis* trophozoites but could not
replicate and were degraded eventually.

## Discussion

This study reports on the discovery of the first virus infecting the
amoeboflagellate *Naegleria*. The giant *Catovirus naegleriensis* (NiV)
represents a sister branch to a metagenomic assembled genome within
the Klosneuviruses (*Mimiviridae*) (Fig. 1)[3]. Like many other mimiviruses, the NiV genome encodes for an increased amount of translation related genes. However, similar to the Bodo saltans virus genome,
no tRNAs but tRNA repair genes were detected[11], indicating that NiV
relies on the tRNA pool of its *Naegleria* hosts.

We observed explicit adaptations in the NiV genome to *Naegleria* hosts. The virus shares 3.1% of its genes with its ameba host. This is a general trend of *Nucleocytoviricota*, which show pronouncedly higher rates of horizontal gene transfer events with their hosts compared to all other viral lineages[37]. This pattern of gene sharing could be caused

by the integration of viral genomes into those of their hosts, which is widespread among giant viruses infecting algae[38]. However, endogenization of viral genomes in *Naegleria* genomes has not been observed to date, and our phylogenetic analysis of the potentially horizontally acquired genes as well as their putative involvement in the manipulation of host cellular pathways, point to rather recent acquisitions of host-derived genes into the NiV genome (Fig. 3, Supplementary Table 2). Accordingly, NiV most likely gained genes encoding a HSP70 and a mitochondrial chaperone, and genes associated with vesicular trafficking, cell cycle, and apoptosis inhibition from a *Naegleria*-like host. The apoptosis inhibition factor is also present in the virion proteome, indicating that the virus might be able to counteract the host's defense through programmed cell death.

Light and electron microscopic studies of the NiV replication cycle in *Naegleria clarki* provided evidence that it resembles the Mimivirus replication cycle in *Acanthamoeba polyphaga*[29,30] despite the large evolutionary distance and different ecology of their protist hosts (Figs. 4–6). The virion enters the host cell via phagocytosis. After the stargate opening, the viral membrane fuses with the phagosomal membrane, releasing the viral core. In the cytoplasm, the viral core gives rise to the viral factory[29]. Couriously, some viral particles appeared to be stuck and degraded in multi-laminar membrane structures (Fig. 6a, d), which could represent a potential host defense mechanism.

One peculiar observation we made was that axenic *Naegleria* cultures seemed to be resistant to infection with NiV, while the host cells died post-viral infection under monoxenic conditions in which *Naegleria* feeds via phagocytosis on bacteria. Giant viruses enter the host cell via different pathways, ranging from phagocytosis over pinocytosis to specific receptor-dependent entry mechanisms[34]. *Mimiviridae* rely on phagocytosis for host entry[34]. They have fibrils that mimic the structure of peptidoglycan[36,39]. Presumably, this mediates recognition as potential food bacteria and triggers phagocytosis by the ameba host[34]. We speculate that axenic *Naegleria*, like other amoebae, switch their feeding mode from phagocytosis to pinocytosis in the absence of bacteria[40], thereby circumventing the entry mechanism of NiV. Similar restrictions might apply to other heterotrophic protists,

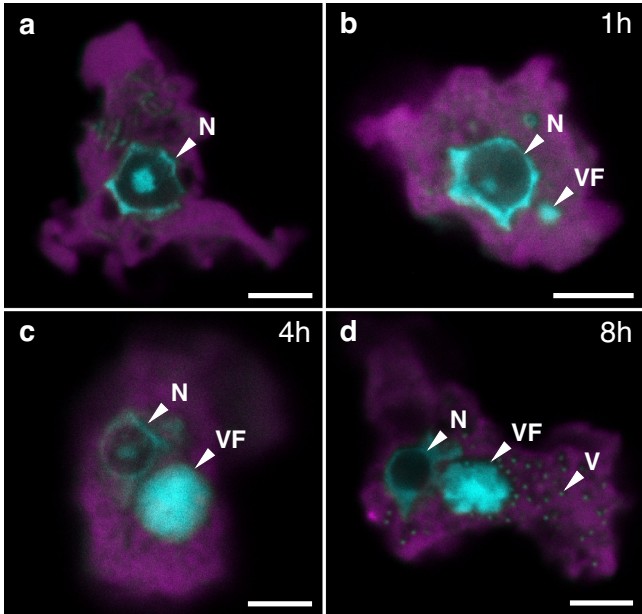

**Fig. 4 | The Naegleriavirus replication cycle in *Naegleria clarki*.** The infection was performed at a multiplicity of infection (MOI) of 10. Fluorescence in situ hybridization (FISH) images are shown. The host cell is depicted in magenta (oligonucleotide probe Nag1088; Supplementary Table 3), with nucleic acid staining by DAPI in cyan. **a** An uninfected *N. clarki* trophozoite. **b** An ameba cell 1 hour post viral infection; small DAPI-stained structures start to accumulate in the cytoplasm. **c** 4 hpi, intermediate stages of the viral factory are visible. **d** 8 hpi, the major viral factory and mature viral particles accumulating in the host cytoplasm can be seen. **N** = nucleus. **VF** = viral factory. **V** = virion. Scale bar = 5 μm.

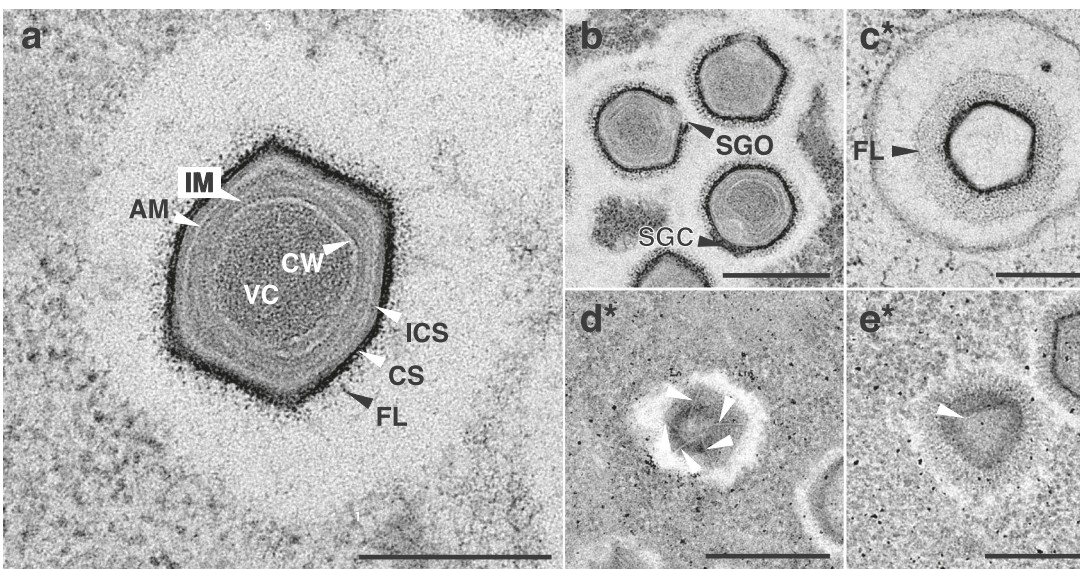

**Fig. 5 | The ultrastructure of Naegleriavirus virions.** Transmission electron microscopy of cryoimmobilized samples processed by freeze substitution or chemically fixed at room temperature (indicated by *). **a** Medial section of a virion in the host cytoplasm exhibiting the viral core surrounded by membranes, the capsid shell, and a fiber shell[31]. The NiV particle structure is notably similar to that of mimiviruses[31]. **b** Medial section of virions showing open and closed stargates (arrowheads). **c** Virion within a host cell phagosome devoid of its viral core, and

membranous structures. Note a well-visible fiber layer covering the electron-dense capsid shell of the virion. **d** Vertex with starfish-shaped edges (arrowheads). **e** A lateral section showing a triangular profile of the icosahedral capsid shell surrounded by the fiber layer. All scale bars: 500 nm. **CS** = capsid shell. **CW** = core wall. **AM** = proposed additional membrane. **FL** = fiber layer. **ICS** = inner capsid shell. **IM** = inner membrane. **VC** = viral core. **SGC** = stargate closed. **SGO** = stargate opened. Terminology is based on[29,30].

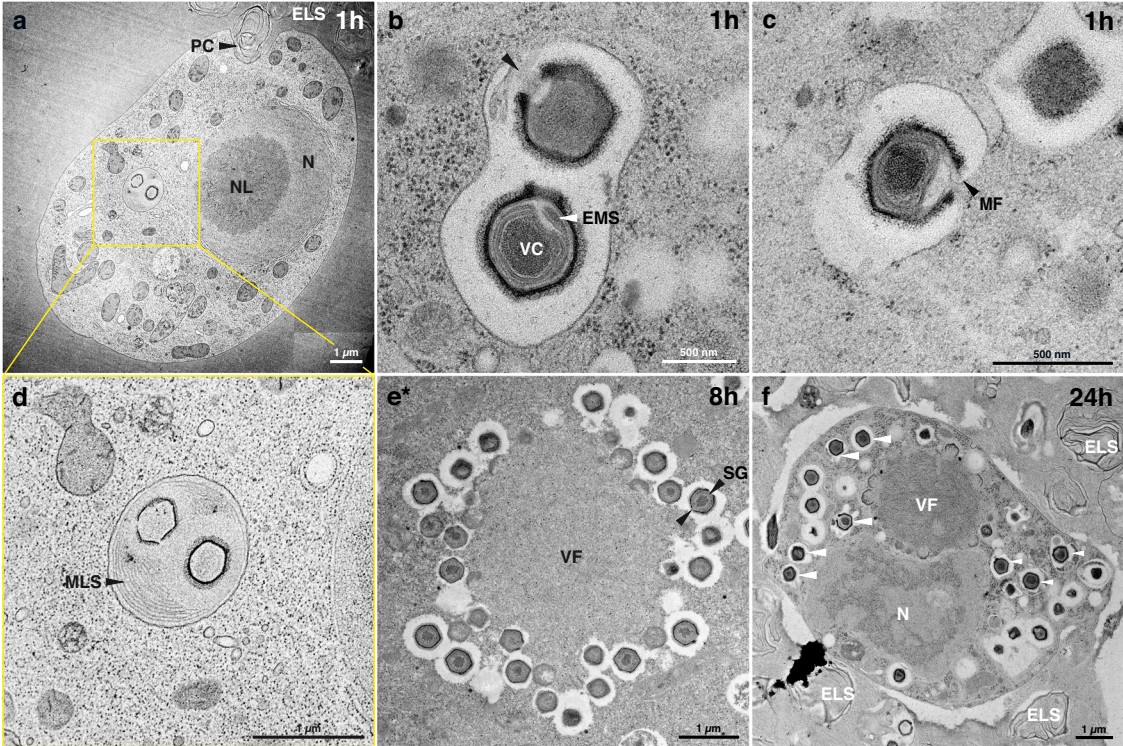

**Fig. 6 | Features of the Naegleriavirus replication cycle.** Transmission electron microscopy of cryoimmobilized samples processed by freeze substitution or chemically fixed at room temperature (indicated by *). Numbers indicate hours post-infection (hpi). **a** *N. clarki* trophozoite forming a phagocytic cup and containing two virions in the phagosome (boxed rectangle); potentially phagocytosed virion marked with an arrowhead. **b** Open stargate (black arrow head) in the upper virion. Virion underneath displays the beginning of the stargate opening; the extra membrane sac can be seen. **c** Fusion of a NiV inner membrane with a phagosome membrane (black arrow head). **d** Two virions devoid of viral cores enclosed by a multilamellar structure within a phagosome. **e** Fully grown and productive viral factory. **f** Infected ameba showing a large viral factory and virions accumulating in the host cytoplasm. Membrane-enclosed virions suggest ongoing phagocytosis (white arrowheads). **ELS** = extracellular lamellar structures. **EMS** = extra membrane sac. **MF** = membrane fusion. **MLS** = multilamellar structure. **N** = nucleus. **NL** = nucleolus. **PC** = phagocytic cup. **SG** = star gate. **VC** = viral core. **VF** = viral factory.

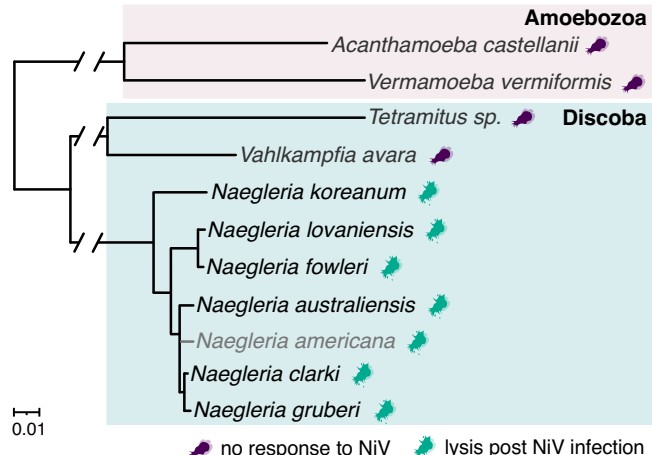

**Fig. 7 | The tested host range of Naegleriavirus.** An 18 S rRNA-based maximum likelihood tree is depicted to visualize the evolutionary relationships of amoebae. The phylogenetic positioning of *Naegleria americana* was inferred from Wang et al.[74] and is shown in gray. The color-coded amoebae indicate whether the host was lysed post NiV infection or not.

highlighting the importance of simulating semi-natural conditions in the lab to isolate giant viruses.

Our findings indicate that NiV is limited to the genus *Naegleria*. We demonstrated that the virus could lyse all tested species within the genus, including the human pathogen *N. fowleri* (Supplementary

Fig. 1). Closely related *Vahlkampfia* species were not susceptible (Fig. 7). Other ameba-infecting giant viruses with known host ranges are also limited to specific genera, species, or strains, respectively, with Tupanvirus as a notable exception[2,4,41,42]. The four available Klosneuviruses isolates, including NiV, infect three distinct host lineages: *Bodo saltans*, *Vermamoeba vermiformis*, and *Naegleria*[11,13,41]. This niche differentiation in evolutionary divergent hosts lends further support for an important role of host switches in the evolution of this viral lineage, as proposed earlier[2,3,41,42].

The temperature range of different *Naegleria* species varies from 20 to 46 °C[36], and we show here that NiV infects or inhibits both mesophilic as well as thermophilic species at ambient and elevated temperatures (Figs. 7, 8). It is tempting to speculate that the increased amount of chaperons functioning in protein (re-)folding encoded in the virus genome and partially present in the NiV virion proteome, facilitates viral replication in this temperature range. However, our data also suggests that 40 °C, the highest temperature tested, impairs viral replication. Whether this is due to a temperature sensitivity of NiV, for instance by obstruction of viral factory formation or virion assembly, or due to host antiviral defense systems, which might function more efficiently at the growth optimum of this heat-loving ameba, remains unknown.

*N. fowleri* as a human pathogen is problematic mainly in human-impacted ecosystems with elevated temperatures, such as swimming pools and rivers receiving power plant cooling water[36]. With the anticipated increase in environmental water temperatures due to the climate crisis, *N. fowleri* abundance in those ecosystems and, thereby, *N. fowleri* infections and PAM cases are expected to increase[36]. The *Naegleria* infecting virus isolate reported here could represent a first

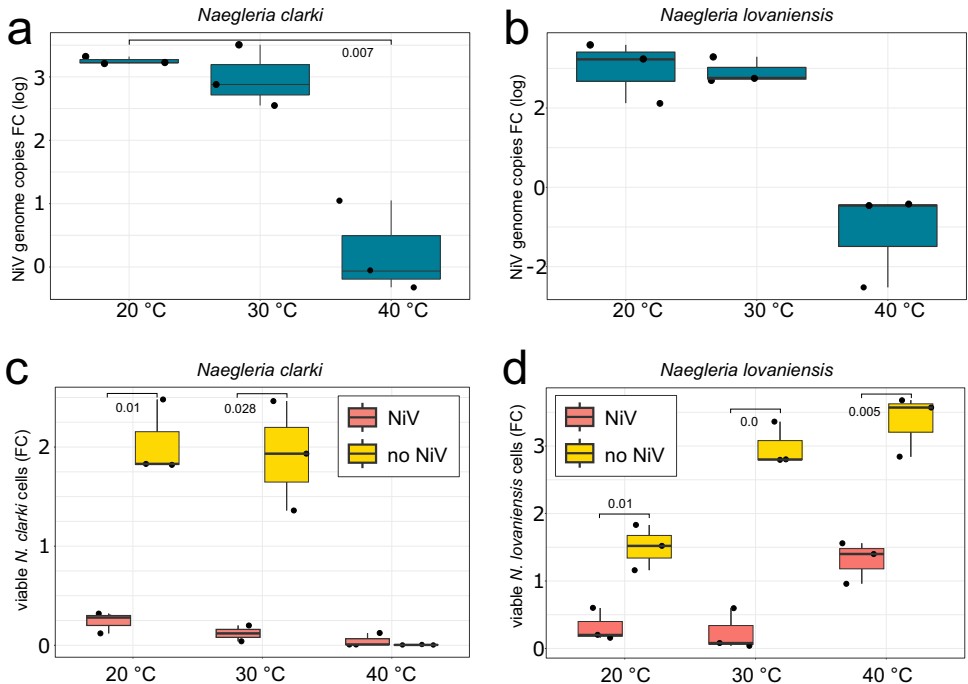

**Fig. 8 | Naegleriavirus replication at different temperatures.** NiV replication at 24 hpi, shown as fold change (FC) of NiV genome copy numbers in *N. clarki* (**a**) and *N. lovaniensis* (**b**) grown at 20, 30, or 40 °C. Fold change of ameba cell numbers of *N. clarki* (**c**) and *N. lovaniensis* (**d**) at 20, 30, and 40 °C with and without NiV at 48 hpi. MOI = 1. Statistical tests compared the viral load 24 hpi for the three setups in (**a**) and (**b**), and the final host concentration at 48 hpi to the initial concentration in (**c**) and (**d**). The (two-tailed) Welch's *t*-test was used for statistical analysis; *n* = 3 independent replicates; *p*-values are depicted (rounded to the third decimal). Boxes indicate the interquartile range, the line indicates the median, whiskers indicate minimum and maximum, respectively. Additional timepoints and ameba cell counts are available as Supplementary Fig. 3.

step towards a viral biocontrol agent for *N. fowleri*. Importantly, it allows us to study processes like programmed cell death and antiviral defense in *N. fowleri*, which will ultimately contribute to a deeper understanding of the biology of this serious human pathogen.

## Methods

### Ameba and *E. coli* cultivation

Amoebae and *E. coli* strain JW5503-1 ΔtolC732::kan were cultivated as described[43]. Briefly, the *Naegleria* species (*Naegleria clarki, Naegleria americana, Naegleria lovaniensis, Naegleria gruberi, Naegleria koreanum, Naegleria australiensis*), *Tetramitus* sp., *Vermamoeba vermiformis, Acanthamoeba castellanii*, and *Vahlkampfia avara* were grown monoxenic in a liquid medium, in 25 cm², 75 cm² or 175 cm² flasks with PAS-buffer and *E. coli* strain JW5503-1 ΔtolC732::kan at 30 °C[44].

The number of amebic cells was counted using a Fast-Read® 102 counting slide (Bioblock Scientific), according to the supplier's recommendations. The counting of at least 5 squares, each square corresponding to a volume of 0.1 μL, was performed for each cell concentration determination. To monitor the viability of the amebic populations, the cells were harvested and centrifuged at 2500 g for 10 min at 20 °C. The pellet was resuspended in 200 μL of PAS buffer by pipetting up and down. 4 μL of the suspension was mixed with 4 μL of commercial 0.4% trypan blue solution and placed on a Fast-Read® counting cell (Bioblock Scientific).

*Naegleria fowleri* ATCC 30808 was grown under axenic conditions in 2% (w/v) Bactocasitone medium, supplemented with 10% (v/v) heat-inactivated fetal bovine serum (FBS), 0.5 mg/mL of streptomycin sulfate (Sigma-Aldrich, Madrid, Spain) and 0.3 μg/mL penicillin G (sodium salt) (Sigma-Aldrich, Madrid, Spain). The trophozoites in the medium were incubated at 37 °C facilitating their multiplication. Subsequently, *N. fowleri* was grown at 32 °C in PAS buffer with *E. coli* strain M15 as a food source. Amoebae were grown in a biosafety level 3 (BSL-3) laboratory facility at the Instituto Universitario de Enfermedades

Tropicales y Salud Pública de Canarias, University of La Laguna, in accordance with the Spanish governments' biosafety guidelines.

### Virus isolation, virion production and temperature experiments

A liter of activated sludge was sampled at the surface of a nitrifying reactor in January 2018 at a wastewater treatment plant in Klosterneuburg (Austria) (48°17'38.8"N 16°20'30.3"E) and transferred into sterile borosilicate bottles. The activated sludge was brought to the laboratory for immediate processing. Viral isolation was carried out as described[43] with the difference that *Naegleria clarki* trophozoites grown monoxenically on *E. coli* strain JW5503-1 ΔtolC732::kan in PAS-buffer were used as a host surrogate instead of an axenic *Acanthamoeba castellanii* culture. Virions were produced as described in[43]. In this study, the multiplicity of infection reflects a multiplicity of particles determined by quantitative PCR. For the experiments with NiV at different temperatures, the amoebae were grown and quantified as described in the section on amoebae cultivation. Assessment of NiV concentration during the temperature time series was carried out with quantitative PCR as described in the section PCR.

### Fluorescence in situ hybridization (FISH)

Fluorescence in situ hybridization (FISH) was performed as described in[43]. Briefly, the cells were collected and transferred to a teflon-coated 10-well slide. After a 45 min adhesion period, cells were fixed with 4% paraformaldehyde for 10 minutes at room temperature. Samples were then dehydrated in ethanol baths (50%, 75%, 96% ethanol for 3 minutes each). On each slide, 10 μl of hybridization buffer (90 mM NaCl, 20 mM Tris-HCl, 0.01% SDS, 25% formamide) and 1 μl of the probes (Supplementary Table 3) Eub338-1-3 labeled with Fluos[45,46], and DAPI at 1 μg/ml were added. The bacterial probe mix Eub338-1-3 was used to detect potential remnants of *E. coli* cells, given that the *Naegleria* was grown monoxenically with *E. coli*. For *Naegleria*, a custom-designed probe Nag1088 (Cy5) was used instead of the Euk516 probe

(Supplementary Table 3). Hybridizations were conducted in moist chambers at 46 °C for 1.5 hours. Slides were washed with 12 µl of washing buffer (1 mM Tris-HCl, 0.25 mM EDTA, and 0.149 M NaCl), followed by a 10-minute incubation in the washing buffer at 46 °C. Slides were then briefly washed in cold water (~4 °C) and air-dried. Finally, the slides were examined using a confocal laser-scanning microscope (Leica SP8).

## Transmission electron microscopy (TEM)

The replication cycle of the NiV was investigated with TEM at 0, 1, 2, 5, 8, 16, and 24 hpi. TEM was carried out as in[43]. Samples were centrifuged for 10 min at 3000 g and chemically fixed in 2.5% glutaraldehyde (Agar Scientific) in 0.05 M phosphate buffer (Merck) for one hour, washed three times for 10 min in the same buffer and post-fixed with 2% aqueous osmium tetroxide (EMS/Science Services) solution for one hour and rewashed three times for 10 min in water (both fixations at RT).

Then the fixed pellets were dehydrated in a series of ethanol (30%, 50%, 70%, 95% for 10 min each; 2 times 100%) for 10 min each, followed by 100% acetone for 5, 7, and 10 min. Subsequently, ethanol was replaced by 100% acetone, followed by infiltration with mixtures of Epon resin (Agar Scientific Ltd., Stansted, UK): 3:1 for 10 min, 3:1 for 60 min, 1:1 for 20 min, and 1:1 for 120 min, followed by the slight opening of the Eppendorf tubes for evaporation of the acetone overnight. The infiltrated samples were placed in fresh resin twice for 1 hour, whereby a desiccator at 250 mbar was used for the second round. Finally, the resin was exchanged for the last time, and the samples were heated in a heating cabinet at 45 °C for 1 hour and polymerized at 60 °C for three days.

For high-pressure freezing, samples were centrifuged for 10 min at 3000 g. Parts of the concentrated pellets were transferred into carriers for high-pressure freezing type A (6 mm in diameter, 200 µm in depth) and covered with the flat surface of carriers type B (Leica Microsystems, Austria). Before use, the carriers were coated with 1-hexadecene (Merck, Sharp). To avoid air inclusions, carrier A was filled up with 20% Bovine serum albumin (BSA) before covering with carrier B. Subsequently. The mounted sample carrier sandwich was frozen at about 2000 bar with the high-pressure freezer HPM100 (Leica Microsystems, Austria). Once released automatically in a Dewar vessel filled with liquid nitrogen, the high-pressure frozen carrier sandwich was separated from the middle plate under liquid nitrogen using a punching device.

Freeze substitution was performed in an automated freeze substitution system AFS2 (Leica Microsystems, Austria) equipped with an agitation module (Cryomodultech e.U., Austria)[47]. Carriers containing the high-pressure frozen samples were placed onto 1 ml liquid-nitrogen frozen 1% OsO$_4$ in acetone in 2 ml- cryotubes. Afterward, the tubes were inserted in the tube holders of the agitation module in the cryo-chamber of the AFS2 (pre-cooled to −140 °C). Freeze substitution occurred under agitation (20 V) at −85 °C for 10 h. Then the samples were warmed up to room temperature. This step required programming considering the temperature difference between the real temperature in the cryotube and the temperature sensor of the AFS2 as described in[47]. The following temperature program was used: on day 1 for 30 min from −140 to −105 °C, 10 h at −105 °C, 30 min from −105 to −90 °C, 2 h at −90 °C, 4 h from −90 to 20 °C and on day 2 for 1 h 20 °C. Samples were infiltrated with 100% acetone and epoxy resin Agar 100 (Agar Scientific Ltd., Stansted, UK) mixtures according to the following schedule: 2:1 for 15 min, 1:1 for 30 min, 1:2 for 150 min. Subsequently, samples were transferred in embedding molds and infiltrated with pure resin overnight. The resin polymerized in the oven at 65 °C for about 36 h.

Ultra-thin sections with a 70−90 nm thickness were produced with Leica ultramicrotomes (EM UC7 and Ultracut S, Leica Microsystems, Austria) using a diamond knife (Diatome, Nidau, Switzerland) and mounted on 200 mesh copper grids. The grids holding the ultra-thin sections were mounted on a grid holder and single drop stained with gadolinium-III-acetate (Sigma Aldrich)[48] for 30 min, rinsed with and immersed in water. Excess water was removed with filter paper. Subsequently, the grids were single drop post-stained with 3% lead citrate for 8 min (Leica Microsystems)[47] (in a covered petri dish with NAOH patties scavenging lead carbonate) and washed again with water. Alternatively, a portion of the sections was stained with 2, 5% uranyl acetate for 10 min followed by 3 % lead citrate for 8 min or 4% Neodymium III acetate (Alfa Aesar, Thermo Fisher) for 30 min instead of gadolinium-III-acetate[48]. The stained ultra-thin sections were then examined with the transmission electron microscope Zeiss Libra 120 with LaB 6 filament, 120 kV, and a bottom mount camera (Sharp: eye TRS 2 x 2k) and ImageSp-professional software (Tröndle, Moorenweis, Germany).

## DNA isolation and quantitative PCR

Isolation of genomic DNA for genome sequencing was carried out with a cetyltrimethylammonium bromide-based extraction method[49]. Viral DNA for PCR was extracted using QIAGEN Blood and Tissue kit® and was automated using the Qiacube connect (Qiagen). The purified DNA served as a template for different PCR reactions, using the DreamTaq polymerase according to the manufacturer's recommendations (Thermo Fisher Scientific).

Quantitative real-time PCR (qPCR) was performed by amplifying an 176 bp fragment of the DNA-dependent DNA polymerase subunit beta gene, using the newly designed primers DNA_pol_NiV_1704F and DNA_pol_NiV_1880R (Supplementary Table 3). The primers were designed with the Primer 3 tool (www.primer3.ut.ee/) and tested in silico with the PCR primer stats tool (www.bioinformatics.org/sms2/pcr_primer_stats.htlm). The master mix Takyon™ No Rox SYBR® Master Mix dTTP Blue kit (Eurogentec) included 5 µl of Takyon Mix (1 X), 0.25 µl of each primer (250 nM), 2.5 µl of nuclease-free water and 2 µl of DNA. The cycling conditions comprised an initial denaturation for 3 min at 95 °C, then 45 cycles of denaturation at 95 °C for 10 s, annealing at 60 °C for 40 s, and a melting curve which consisted of 5 s at 95 °C then 1 min at 65 °C and finally a continuous increase in temperature to 95 °C with 5 fluorescence acquisitions per °C. Gene copy numbers and qPCR efficiencies were calculated using standard curves for each experiment. An 840 bp long NiV DNA-dependent DNA polymerase subunit beta fragment was amplified with the primer pair DNA_pol_NiV_1512F and DNA_pol_NiV_2352R (Supplementary Table 3), which served to establish a standard curve for both efficiency calculation and absolute quantification. qPCR efficiency with the above-mentioned primers was estimated at 100.5%. Amplification was carried out with an initial denaturation step of 2 min at 95 °C, followed by 35 cycles of denaturation 45 s at 95 °C, hybridization 30 s at 57 °C, and elongation 120 s at 72 °C. A final elongation step for 5 min 72 °C was performed. After amplification, PCR products were checked on agarose gel (1.5% agarose, 0.5 x Tris Borate EDTA, 2.5 µL Midori Green), 100 V for 30 min. The concentration of the purified PCR product was determined with NanoDrop™. All standard curves corresponding to qPCR experiments presented in this manuscript had $R^2$ values higher than 0.99.

## Genome sequencing, annotation, and analysis

The NiV genome was sequenced using Illumina (MiSeq-based platform) and PacBio sequencing platforms, respectively. The quality of the Illumina reads was checked visually with FastQC (version 0.11.9, https://www.bioinformatics.babraham.ac.uk/projects/fastqc). Artifacts filtering was done with fastx_artifacts_filter (default parameters) from the FASTX-Toolkit (version 0.0.13, http://hannonlab.cshl.edu/fastx_toolkit). Trimming on the right tail of the reads by the quality and minimum length was done with fastq_quality_trimmer with a minimum quality threshold of 20 and a minimum read length of 75 base pairs (bp). To perform the last stage of filtering of exact duplicate reads,

removal of unknown nucleotides (N), and left tail trimming, we utilized prinseq-lite.pl (version 0.20.4, http://prinseq.sourceforge.net) (Schmieder and Edwards 2011). During this process, reads containing Ns were filtered out, and trimming from the left end of the reads was performed, considering a minimum quality score of 20. We removed exact duplicates and reverse complement 5′/3′ duplicates while enforcing a minimum read length of 75 bp. A random subsample with 3% of the PacBio reads was generated and subsequently filtered and trimmed with canu (version 2, https://github.com/marbl/canu)[50] using default parameters. A hybrid assembly comprising the trimmed and filtered forward and reverse Illumina reads and the trimmed sub-sampled PacBio reads were made using Unicycler (version 0.4.8, https://github.com/rrwick/Unicycler)[51] in normal mode.

Open reading frame (ORF) prediction and a first annotation were carried out with PROKKA (version 1.14.6)[51,52], tRNAs and other non-coding RNAs were searched for with tRNAscan-SE (version 2.0.9)[53], and Infernal[54], respectively. GC content and skew were calculated with GCcalc.py (https://github.com/WenchaoLin/GCcalc). The annotation was manually curated and augmented with Unipro Ugene (version 37.1)[55] using the results of a functional prediction and domains search with InterProScan (version 5.45-80.0)[56]. To gain additional information about the functions of the predicted genes, a blastp (BLASTP, version 2.10.0) search against a local monthly mirror of the NCBI nr database (February 2021, -evalue 0.00001), a search against precomputed orthologous groups with eggNOG-mapper v2 (http://eggnog-mapper.embl.de) against eggNOG 5.0[57], and a BlastKOALA search against KEGG with default options[58] were performed. For the identification of intron sequences, we used cmscan within Infernal v1.1.4[54] and the Rfam models RF00028.cm, RF01999.cm, RF02005.cm, RF00029.cm, RF02001.cm, RF02012.cm, RF01909.cm, RF02003.cm, RF02266.cm, RF01998.cm, RF02004.cm. Using this approach we successfully detected introns in some giant viruses, including Bodo saltans virus, but we failed to detect any self-splicing introns in NiV. Properties of the Naegleriavirus genome and bidirectional best BLASTP hits between *Catovirus naegleriensis*s and *Catovirus CTV1* were visualized using Circos[58,59]. Remote homology searches were performed with HHpred[21].

## Proteome analysis

After centrifugation for 15 min at 12,000 g and 4 °C and discarding the supernatant, the pellet was washed twice with 1.8 ml ice-cold 100% MeOH and ultrasonicated for 5 min, centrifuged for 10 min at 12,000 g and 4 °C. This washing step was repeated once with 1.8 ml ice-cold acetone, followed by centrifugation for 10 min at 14,800 g and 4 °C. After discarding the supernatant, the pellet was left to dry under the hood. The dried pellet was suspended in 40 μl urea extraction buffer (8.8 M Urea and 500 mM Hepes in a ratio of 9:1) and put for 30 min on a shaker at 700 rpm at RT. After centrifugation for 5 min at 14,800 rpm and RT, the supernatant (dissolved proteins) was transferred to a 0.5 LoBind tube, and the pellet (mainly salts) was discarded. In order to determine the protein concentration in the sample, a Bradford assay against bovine serum albumin gradient was performed[60]. In order to reduce the thiol groups of cysteines, the sample was adjusted to 5 mM dithiothreitol (DTT) and incubated for 30 min on a thermoshaker at 37 °C and 500 rpm. Then the sample was alkylated on RT by adjusting to 10 mM iodoacetamide (IAA) and incubated in the dark for 30 min on a thermoshaker at 23 °C and 500 rpm. The alkylation was stopped by adjusting the sample to a total of 10 mM DTT. For digestion, 3 sample volumes of 13.3% acetonitrile (ACN), 33.3 mM ammonium bicarbonate, 13.3 mM $CaCl_2$, and 3 μl of trypsin beads (Promega, Germany) were added. The sample was incubated at 37 °C rotating at 10 rpm overnight. Desalting was performed using C18-Bond Elut 96-well plates (Agilent Technologies, United States) connected to a water jet pump. The plate was activated twice with 800 μl MeOH and washed twice with 800 μl water. The sample was loaded in the middle of the C18 membrane and incubated for 10 min at RT without the water jet pump on.

The peptides were subsequently washed twice with 800 μl water and eluted with $3 \times 800$ μl MeOH. The peptide liquid was then dried in a vacuum concentrator and stored at −80 °C.

Liquid chromatography coupled to tandem mass spectrometry: The sample was resuspended in 2% ACN (0.1% FA) solution, and the final protein concentration was adjusted to 50 ng/μl. 1 μg of the sample was injected for the measurement, which was performed with liquid chromatography coupled to tandem mass spectrometry, LC-Qexactive-Orbitrap-MS (HPLC UltiMate 3000 and MS/MS Qexactive plus, Thermo Fisher Scientific). Measurement settings: Full scan range 350–1800 m/z resolution 1,20,000 max. 20 MS2 scans (activation type CID), repeat count 1, repeat duration 30 sec, exclusion list size 500, exclusion duration 30 sec, charge state screening enabled with a rejection of unassigned and +1 charge states, minimum signal threshold 1000[61].

Proteins were identified using a NiV proteome fasta file, and since there was no annotated genome of the host *N. clarki* available, the proteome file of the close relative *N. gruberi* was retrieved from uniprot (up000006671) and used for the subsequent analysis. MaxQuant v1.6.5.0 with the following standard parameters was used: first search peptide tolerance 20 ppm, main search tolerance 4.5 ppm, ITMS MS/MS match tolerance 0.6 Da. A maximum of three of the following two variable modifications were allowed per peptide: oxidation of methionine and acetylation of the N-term. The fixed modification was carbamidomethylation of cysteine. A maximum of two dismissed cleavages were tolerated. Identifications were matched between runs in a 0.7 min window. An FDR cutout at 0.01 (at Peptide Spectrum Match and protein level) was set with a reversed decoy database. A minimum of seven amino acids was required for the identification of peptides. Identifications were separated into those with at least two distinct peptides and those identified by only one peptide but detected in each out of the five replicates analyzed (Supplementary data set 3).

## Phylogenetic analyzes

The data set from Guglielmini et al.[62] containing alignments of core proteins for up to 61 viral species, was used as a basis for the phylogenetic placement of NiV. The respective amino acid sequences of additional members of the Klosneuvirinae, Bodo saltans virus, Yasmi-nevirus, and Fadolivirus[41], and NiV were added to the dataset. Three almost universal NCLDV core proteins, DNA polymerase family B, D5-like helicase primase, and poxvirus late transcription factor VLTF3[63] were used for phylogenetic analysis and aligned individually using MAFFT-linsi version 7.520[63,64]. The alignments were then filtered with trimAl (version 1.41) and the parameter -automated1 for a heuristic selection of the optimal trimming method based on similarity statistics[65]. The filtered alignments were subsequently concatenated using a perl script (https://github.com/nylander/catfasta2phyml). Intermediate visual checks of the alignments were performed with SeaView (version 5.05)[66]. The concatenated alignment was used to calculate maximum likelihood (ML) trees with IQ-TREE 2 (version 2.2.7) using a mixture model (LG + F + I + R6), SH-aLRT test and ultrafast bootstrap with 1000 replicates (-B 1000 -alrt 1000)[67–69]. Trees were visualized with iTOL (version 6.7.6)[70].

The phylogeny of the ameba species was reconstructed using 18 S rRNA genes. The sequences were imported into the Seaview software (V4.7)[66] and then aligned with the MUSCLE (V3.8.31)[71]. The phylogeny was inferred using the maximum likelihood method as implemented in IQ-Tree 2. Finally, 1000 bootstrap iterations were performed to test the robustness of the resulting nodes.

To investigate potential horizontal gene transfer events between NiV and protist hosts, all NiV genes with emapper hits to Eukaryota were collected. Out of those genes, all genes with hits to genes from protists were manually selected. For each of those, all available sequences of the corresponding protein family were collected from

the eggNOG 5.0[57] database and supplemented with viral homologs (BLASTp) if available. Sequences were clustered and dereplicated with CD-HIT (V4.8.1, settings: -c 0.8 -M 8000)[72], followed by an alignment of cluster representatives with MAFFT (V7.520, default settings)[63,64], a trimming step with trimal (V1.4.rev15, -gt 0.2)[65], and finally, tree construction with fast tree (version 2.1.11, default settings)[73].

## Statistics and reproducibility

Data visualization and statistical tests were carried out using R-studio (https://www.rstudio.com/, version 4.2.1). Growth and infection experiments were repeated at least three times. FISH and TEM images in Figs. 4, 5, and 6 are representative images from at least two independent experiments.

## Reporting summary

Further information on research design is available in the Nature Portfolio Reporting Summary linked to this article.

## Data availability

Sequencing data and the annotated Naegleriavirus genome generated in this study have been deposited in Genbank/ENA/DDBJ under accession code OZ003748 and as BioProject PRJEB72011. Proteome data has been deposited at ProteomeXchange under accession code PXD044853. Supplementary phylogenetic tree files are available at figshare under the https://doi.org/10.6084/m9.figshare.25117901. Source data for Fig. 8 and Supplementary Fig. 3 are provided in this paper. Source data are provided in this paper.

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

## Acknowledgements

We thank Martina Michalikova for valuable feedback during the writing of the manuscript and Bela Hausmann sequencing and for help with bioinformatics and R. The Life Science Compute Cluster at the University of Vienna (LiSC; https://lisc.univie.ac.at/) was used for computational analysis. We thank our funders: Austrian Science Fund (doc. funds project MAINTAIN, DOC 69-B to MH; project P37198-B to MH), European Union (IF GIVIREVOL 891572 and ERC CHIMERA 101039843 grants to AW), Region of Nouvelle Aquitaine ("Habisan program" CPER-FEDER to VD), Consorcio Centro de Investigación Biomédica de Enfermedades Infecciosas (CIBERINFEC), Instituto de Salud Carlos III (CB21/13/00100), Cabildo Insular de Tenerife 2023–2028, Ministerio de Sanidad, Spain. The work conducted by the U.S. Department of Energy Joint Genome Institute (https://ror.org/04xm1d337), a DOE Office of Science User Facility, is supported by the Office of Science of the U.S. Department of Energy operated under contract number DE-AC02-05CH11231. Views and opinions expressed are, however, those of the author(s) only and do not necessarily reflect those of the European Union or the European Research Council Executive Agency. Neither the European Union nor the granting authority can be held responsible for them. This research was

funded in part by the Austrian Science Fund (FWF) [DOI 10.55776/DOC69]. For the purpose of open access, the author has applied a CC BY public copyright license to any author accepted manuscript version arising from this submission.

## Author contributions

P.A., V.D. and M.H. conceptualized this study. P.A. isolated the Naegleriavirus. P.A., F.P., A.H. and V.D. carried out all experiments involving the virus and ameba other than *N. fowleri,* which were performed by I.S., I.A.J. and J.L.M. Electron microscopy was performed by P.A., N.C., F.P. and S.R. Proteomics analysis was performed by P.A. and S.W. Genome sequencing, analysis, and interpretation was performed by P.A., F.P., A.W., F.S. and M.H. The first manuscript version was written by P.A. and revised by P.A., F.P. and M.H. All authors commented on or edited the manuscript.

## Competing interests

S.R. acts as advisor for Cryomodultech, e.U., companies registered in Vienna No. FN460571. All other authors declare no competing interests.
