## [Peer Review File · Nature Communications]

A giant virus infecting the amoeboflagellate NaegleriaREVIEWER COMMENTS

Reviewer #1 (Remarks to the Author):

Arthofer et al present a new giant virus from the family Mimiviridae, which infects the amoeboflagellate *Naegleria* and thus differs from the usual *Acanthamoeba*-infecting mimiviruses. It is also interesting because it represents the first isolate from the waste water treatment plant in Austria where the first Klosneuvirus was found by metagenomics. The authors thus follow up on previous findings of a very interesting giant virus genome and managed to isolate a related giant virus in culture, which is a commendable feat. The virus displays similarities to mimivirus in its infection cycle, but differs genomically, mostly by several host-derived genes. This virus thus nicely highlights the adaptability of giant viruses. The field of environmental virology will surely benefit from this discovery and from future experimental studies on this interesting system. The paper is well written, although some issues need to be clarified, as outlined below.

- Introduction, l 54-59: This first paragraph could be re-written, as the first few sentences are not logically linked and don't contribute to necessary information for understanding the paper.
- Fig. 1A: Is GC skew more informative here than GC content? If not, GC percentage should be shown instead.
- l 102: How are positive and negative strands defined in giant viruses? For NiV, was this adopted from the orientation of the closest relatives (Cotovirus), or is it based on the strand location of certain marker genes? There is probably no information available about the mechanics of DNA replication, which would allow for an unambiguous definition? If not, the classification in positive and negative strands seems rather random.
- Fig. 2: Description for 2C is missing in the legend. The color schemes should be better separated between the set for 2A and the other set for 2B+C. Also, the blue shades are easily confused and should be replaced (e.g. orange, brown).
- l. 125, 216, 275: "Mimiviridae" is a taxon name and should not be used to refer to virus properties. See <https://link.springer.com/article/10.1007/s00705-021-05323-4>. Thus, the correct phrasing would be "All virion key proteins of members of the family Mimiviridae...", or "All virion key proteins of mimiviruses..."
- l. 130: So far, all giant virus isolates have one major capsid protein, and a variable number of capsid protein homologs. The MCP is the capsid protein with the highest abundance in the virion protein and constitutes the faces of the icosahedral capsid. The other capsid proteins are often used for unique capsid features, e.g. as anchor points for fiber attachments etc. An even better description includes the type of capsid protein fold, here the DJR fold. Here, CDS 706 appears to be the DJR-MCP, whereas the others may have been too low copy numbers to be detected. Were homologs of the putative penton and tape measure proteins found in NiV (see PMIDs 33074779, 36309542)?
- l. 158-163: Can you provide more details about the repetitive regions of the NiV genome? It seems from Fig. 1A that there are far more paralogs present than the ones mentioned in this paragraph.
- l. 169-177: Are these repeat regions located within a predicted protein-coding region? Is there detectable sequence or functional similarity of this or nearby CDSs to the MIMIVIRE-associated proteins from mimivirus?
- Fig. 4: I am confused about the FISH probes used here. The legend does not specify any probes and their colors, instead it appears that only DAPI staining was used? The methods state that FISH was used with two probes, one bacterial (why?) and one amoebal (used where?).

- Fig. 5: "IM" is barely readable. By "extra membrane", do you mean external/outer membrane? Otherwise, how do you know this the extra membrane, and not the inner one? How do you differentiate between membranes and protein layers? "Fiber layer" is probably a better term than "Fiber shell", as shell implies a continuous surface.
- Fig. 6: The area in D comprises more than the box in A indicates. A higher magnification for D would be good. Can the contrast of A be increased?
- I. 228: You do not actually show the sequence of events that leads to the stargate opening. This is of course the logical assumption by comparison with mimivirus studies, but still an invalid conclusion from the experiments done here. Please re-phrase.
- I. 236/237: How many cells with multilamellar phagosomes did you examine? Are these conclusions based on single cross-sections, or tomograms comprising the entire phagosome?
- I. 241: If there are mitochondria in Fig. 6E, they should be clearly marked. Even then, I cannot see that mitochondria would be overrepresented around the virion factory compared to other cell parts.
- Suppl. Fig. 2: Not all abbreviations are mentioned in the legend.
- I. 242: Could the extracellular laminar structures also have been produced by bacteria in the culture, or result from lysed amoebae?
- I. 244: Do cells in uninfected cultures form cysts?
- Fig. 7: A better way to test for host range is to monitor viral DNA replication, or production of virions in combination with lysis. Lysis as the sole criterion may miss low levels of virion production and release.
- I. 268: Why was this finding curious? Should it not be expected that there is no virus production if the host population does not decline? A simple experiment would help to clarify the role of temperature on NiV infectivity: incubate an aliquot of cell-free virus at 20, 30, or 40 degrees C for 30 min, then infect cultures at ambient temperature with these treated viruses to see if they can establish a productive infection.
- I. 276: Bodo saltans virus is not italicized, as you refer here to the virus, not the taxon.
- I. 292/293: "viral escape mechanism" is used here in the sense of host defense, but could easily be misunderstood in the sense of virions exiting the host cell.
- I. 296-307: very nice interpretation, I completely agree!
- I. 326: *N. fowleri* is dangerous to humans because it enters the brain and causes infections there, destroying brain tissue. I fail to see how giant viruses could be used as a treatment, because it would surely be undesirable and also impractical to deliver the virus to the brain. Using the virus in a pre-emptive manner would mean producing large quantities of the virus and treating immense volumes of water on a regular basis with this virus. I do not think that the number of cases justifies such intense and costly measures. Please correct me if I overlooked an important aspect here, but given this brief assessment, I would not emphasize the potential use of NiV as a biocontrol agent, as it may give false hope and seems quite far-fetched. Better concentrate on the interesting virological and evolutionary aspects of your study, which are outstanding by themselves!
- Data availability: the genome accession number of NiV should be included in the final version of this paper.
- Suppl. Fig. 1 legend: replace *Catovirus naegleriensis* (taxon name, has no genome) with Naegleriavirus (trivial name, has a genome)
- As there are no supplementary tables 1+2, rename S. tables 3+4 to 1+2.

Reviewer #2 (Remarks to the Author):

Arthofer et al present a study describing the isolation and characterization of a giant virus that infects hosts in the genus *Naegleria*. This is a well-written study and the figures are clear. The genomic, proteomic, and microscopy-based analyses are comprehensive. It is exciting that a virus infecting the genus *Naegleria* has been found (especially considering it infects the infamous *Naegleria fowleri*). This will be of great interest to those studying giant viruses as well as those who work with protist pathogens, and so it is important that this study is published in a journal with broad readership. The discovery of temperature-dependent virulence is also quite interesting - I have not seen this shown for other systems.

I only have some minor suggestions that may help to improve the manuscript:

Phylum names should be italicized as per ICTV recommendations <https://ictv.global/faqs>

Line 104- I am pleased to say that the subfamily Klosneuvirinae has been officially ratified by the ICTV, so “proposed” can be removed.

Line 135 - the authors may wish to follow up with an hhpred search - this is more sensitive than interpro and may help confirm that these proteins are mCPs

Lines 154-164: the discussion of viral dynamin could include a recent experimental study that shed light on the function of these viral proteins. This study indicated that viral dynamin was involved in remodeling mitochondrial membranes during infection (<https://doi.org/10.1093/molbev/msad134>). Was any obvious changes in the mitochondria apparent during infection here? In addition, the Khalifeh et al study on viral SNAREs, which the authors correctly cite here, postulates that the patchy distribution of these proteins in viruses is indicative of a virus-host antagonistic evolution. In that light, it may be interesting to note whether these proteins are shared with other members of the Klosneuvirinae, or whether they are specific to this genome. Perhaps this could be included in Panel 1D?

Were any self-splicing introns found? This is a prominent feature of *Bodo saltans* virus.

Line 323 - the temperature sensitivity of the virus is quite fascinating. Is higher temperature linked to any alterations in cellular organellar structures? I wonder if high temperatures could interfere with budding of the viral inner membrane from the cellular ER, for example. I find it somewhat harder to believe that host antiviral systems would be upregulated at higher temperatures, though I could be wrong.

Line 280: “The virus shares 3.1 % of its genes with its amoeba host” This is an interesting pattern and more details here would be useful. One possibility is that this gene sharing is due to a degraded GEVE present in the host genome. GEVEs are quite common in protist genomes and could explain this pattern (<https://doi.org/10.1038/s41586-020-2924-2>). An alternative explanation would be that the virus acquired these genes from the host, in which case it would be interesting to know of any functional predictions here. Some examples of this are shown in Figure 3, but I wasn’t sure if this was the predominant trend or just a few examples. Host genes recently acquired by the virus could be important for specific host-virus interactions and may shed light on the rather narrow host range.

For Figure 2, would it not make sense to include a functional category for structural proteins? I was expecting that for the virion proteome part.

Small point for Figure 3- the authors may wish to increase the line width on the trees, because the thin lines don't show up well on some screens.

Figure 7 is somewhat confusing to me given the vast phylogenetic distances between Naegleria (Discoba) and the other protist groups (Amoebozoa) - perhaps the higher-order classifications should be provided here, perhaps as a bar on the right, or put in the text so it is clear that these are completely different phyla/supergroups?

The phylogenetic methods are robust and I am convinced about the placement, so nothing needs to be re-done here, but in the future the authors may wish to know that there are benchmarked workflows available for making multi-locus trees for giant viruses (see https://github.com/faylward/ncldv_markersearch, described in <https://doi.org/10.1371/journal.pbio.3001430>)

Response to Reviewer Comments

NCOMMS-23-32250

We were excited to see the very positive feedback on our manuscript and would like to thank both reviewers for their valuable and constructive comments. We have addressed all comments and modified the manuscript accordingly.

COMMENTS OF REVIEWER #1

Arthofer et al present a new giant virus from the family Mimiviridae, which infects the amoeboflagellate *Naegleria* and thus differs from the usual *Acanthamoeba*-infecting mimiviruses. It is also interesting because it represents the first isolate from the waste water treatment plant in Austria where the first Klosneuvirus was found by metagenomics. The authors thus follow up on previous findings of a very interesting giant virus genome and managed to isolate a related giant virus in culture, which is a commendable feat. The virus displays similarities to mimivirus in its infection cycle, but differs genomically, mostly by several host-derived genes. This virus thus nicely highlights the adaptability of giant viruses. The field of environmental virology will surely benefit from this discovery and from future experimental studies on this interesting system.

The paper is well written, although some issues need to be clarified, as outlined below.

- Introduction, l 54-59: This first paragraph could be re-written, as the first few sentences are not logically linked and don't contribute to necessary information for understanding the paper.

We thank the reviewer for this suggestion and have modified the first paragraph and a few other bits of the introduction section accordingly.

- Fig. 1A: Is GC skew more informative here than GC content? If not, GC percentage should be shown instead.

Asymmetry in the GC skew can indicate the origin of replication, as e.g. in phaeoviruses (<https://pubmed.ncbi.nlm.nih.gov/19054537/>). Yet, we agree that the GC skew is not informative in the case of NiV. Instead of the GC skew we thus indicate the location of putatively host-derived genes across the genome in the revised Figure 1a.

- l 102: How are positive and negative strands defined in giant viruses? For NiV, was this adopted from the orientation of the closest relatives (*Catovirus*), or is it based on the strand location of certain marker genes? There is probably no information available about the mechanics of DNA replication, which would allow for an unambiguous definition? If not, the classification in positive and negative strands seems rather random.

We agree that there is no clear definition for the distinction of negative and positive strands in this case. We have thus changed the text accordingly: *“The complete, linear double-stranded DNA genome [...] contains 1,000 predicted coding sequences (CDS), 580 on one and 520 CDS on the other strand, respectively.”*

- Fig. 2: Description for 2C is missing in the legend. The color schemes should be better separated between the set for 2A and the other set for 2B+C. Also, the blue shades are easily confused and should be replaced (e.g. orange, brown).

Done as suggested. New and different color schemes are used for the revised version of Figure 2A and 2BC, respectively.

- I. 125, 216, 275: “Mimiviridae” is a taxon name and should not be used to refer to virus properties. See <https://link.springer.com/article/10.1007/s00705-021-05323-4>. Thus, the correct phrasing would be “All virion key proteins of members of the family Mimiviridae...”, or “All virion key proteins of mimiviruses...”

Thank you for noting this mistake. The corresponding sections were adjusted accordingly.

- I. 130: So far, all giant virus isolates have one major capsid protein, and a variable number of capsid protein homologs. The MCP is the capsid protein with the highest abundance in the virion protein and constitutes the faces of the icosahedral capsid. The other capsid proteins are often used for unique capsid features, e.g. as anchor points for fiber attachments etc. An even better description includes the type of capsid protein fold, here the DJR fold. Here, CDS 706 appears to be the DJR-MCP, whereas the others may have been too low copy numbers to be detected. Were homologs of the putative penton and tape measure proteins found in NiV (see PMIDs 33074779, 36309542)?

Thank you for this valuable input. As per suggestion of reviewer 2 we performed additional remote homology searches using HHpred, which confirmed the annotation of the two minor capsid proteins and revealed similarities to penton and tape measure proteins, respectively. We have included these new results and have re-written this paragraph, also to better explain the identification of CDS 706 as the MCP of NiV: *“The NiV genome encodes four major capsid domain-containing proteins (CDS 686, 687, 688, and 706) and one major core protein 4b (CDS 720). In addition, two genes were identified as minor capsid proteins using remote homology searches, which suggest they represent a tape measure protein (CDS 824) and a penton protein (CDS 654), respectively. In the virion proteome, only one major capsid domain-containing protein (CDS 706) was abundantly found, thus likely representing the major capsid protein (MCP) of NiV. Its homologs might represent low abundant proteins involved in specific capsid features.”*

- I. 158-163: Can you provide more details about the repetitive regions of the NiV genome? It seems from Fig. 1A that there are far more paralogs present than the ones mentioned in this paragraph.

These regions mainly contain hypothetical proteins, mostly without any known conserved domains. Sticking out are mainly those highlighted in the text, which include ankyrin and leucine rich repeat domains putatively involved in host cellular pathway manipulation.

- I. 169-177: Are these repeat regions located within a predicted protein-coding region? Is there detectable sequence or functional similarity of this or nearby CDSs to the MIMIVIRE-associated proteins from mimivirus?

The repeat regions are located in intergenic regions. No known MIMIVIRE-associated genes were identified in the vicinity of these repeat regions. This is now better explained in the revised paragraph: *“The NiV genome contains two repeat regions that were not predicted as protein-coding [...]. Notably, we observed no sequence similarity to known MIMIVIRE elements or sequences from known virophages. We also failed to detect protein-coding genes similar to known MIMIVIRE-associated genes in the immediate neighborhood of the repeat regions.”*

- Fig. 4: I am confused about the FISH probes used here. The legend does not specify any probes and their colors, instead it appears that only DAPI staining was used? The methods state that FISH was used with two probes, one bacterial (why?) and one amoebal (used where?).

Thank you for pointing out the missing details on the FISH probes used. Indeed, probe Nag1088 (labeled with Cy5) was used to stain amoeba cells. In addition, a bacterial probe mix (Eub338 1-3 labeled with FLUOS) was used to detect potential remnants of *E. coli* cells as the *Naegleria* cultures were grown monoxenically on *E. coli*. We added this information to both to the figure legend and the respective methods section.

- Fig. 5: “IM” is barely readable. By “extra membrane”, do you mean external/outer membrane? Otherwise, how do you know this the extra membrane, and not the inner one? How do you differentiate between membranes and protein layers? “Fiber layer” is probably a better term than “Fiber shell”, as shell implies a continuous surface.

We have modified the figure to increase readability of the labels.

The reviewer is correct, although we can clearly identify different layers in NiV particles in our analysis, we cannot differentiate between protein and membrane layers. Because of the striking similarities to the structure of Mimivirus particles, we refer to the different layers observed in NiV particles using the terms used for the structure of mimiviruses. This is now better explained in the manuscript text and the figure legend: *“The core wall is surrounded by five distinct layers separated by less electron-dense interstices (Figure 5a). Analogous to previous observations for Mimivirus particles (Mutsafi et al. 2014), these represent the inner membrane, the proposed additional membrane, the inner capsid shell, and the electron-dense hexagon-shaped capsid shell covered by the fiber layer.”*

- Fig. 6: The area in D comprises more than the box in A indicates. A higher magnification for D would be good. Can the contrast of A be increased?

We now provide a higher magnification in panel D, which reflects the area marked by the rectangle in panel A. We also increased the contrast in panel A.

- I. 228: You do not actually show the sequence of events that leads to the stargate opening. This is of course the logical assumption by comparison with mimivirus studies, but still an invalid conclusion from the experiments done here. Please re-phrase.

We modified the corresponding statements, which now read: "*NiV* virions, similar to other giant viruses infecting amoebae, appear to enter the host cell through phagocytosis. [...] We observed closed and open stargates, so we assume that similar to other mimiviruses, the *NiV* stargate changes its conformation from closed to open (Figure 5b)."

- I. 236/237: How many cells with multilamellar phagosomes did you examine? Are these conclusions based on single cross-sections, or tomograms comprising the entire phagosome?

As multilamellar structures were observed only occasionally and as we were not able to quantify these events, we now just state that multilamellar phagosomes were seen sometimes: "*After uptake, the viral particles are located in the phagosome, sometimes surrounded by multilamellar structures (Figure 6a,d).*"

- I. 241: If there are mitochondria in Fig. 6E, they should be clearly marked. Even then, I cannot see that mitochondria would be overrepresented around the virion factory compared to other cell parts.

Agreed, the mitochondria are hardly visible in this figure. As we didn't quantify the distribution of mitochondria during the replication cycle, we omitted the sentence about mitochondria.

- Suppl. Fig. 2: Not all abbreviations are mentioned in the legend.

We added the missing abbreviations.

- I. 242: Could the extracellular lamellar structures also have been produced by bacteria in the culture, or result from lysed amoebae?

We are confident that the extracellular lamellar structures were produced by amoeba as we carefully removed bacterial cells before processing of samples for TEM analysis. Importantly, heterolobosean amoeba are known to produce these structures (e.g. <https://pubmed.ncbi.nlm.nih.gov/33166717/>). However, these structures could indeed be

derived from lysed amoeba. This is now stated in the legend of Supplementary Fig. 2: *“Extracellular multilamellar structures could either be produced actively by the trophozoites or alternatively result from cell lysis.”*

- I. 244: Do cells in uninfected cultures form cysts?

Yes, amoeba generally form cysts under unfavorable conditions. The *Naegleria* species used in this study encysted when incubated in buffer without food bacteria within 12 hours. The infection experiments, however, were carried out under monoxenic conditions with bacteria as food source. In contrast to cultures challenged with NiV, uninfected cultures did not form cysts during the observation time period.

- Fig. 7: A better way to test for host range is to monitor viral DNA replication, or production of virions in combination with lysis. Lysis as the sole criterion may miss low levels of virion production and release.

Agreed, indeed with this method we cannot exclude low levels of virion production. We have added a sentence to clarify this: *“However, as we focused on host cytopathic effects in our experiments, we cannot exclude low levels of virion production without host lysis.”*

- I. 268: Why was this finding curious? Should it not be expected that there is no virus production if the host population does not decline?

A simple experiment would help to clarify the role of temperature on NiV infectivity: incubate an aliquot of cell-free virus at 20, 30, or 40 degrees C for 30 min, then infect cultures at ambient temperature with these treated viruses to see if they can establish a productive infection.

We thank the reviewer for pointing out that this section on the influence of temperature on NiV replication needed additional explanation. In the revised manuscript, we included additional data from the temperature infection experiments, which were performed as time series experiments over 48 h (as the new Supplementary Fig. 3), and modified the legend and manuscript text to better describe the results and enhance clarity.

The reviewer suggested testing temperature sensitivity of the virus by including a pre-incubation of virus particles for 30 min at different temperatures before the infection of the amoeba cultures (at ambient temperature). Indeed, short-term temperature treatment of bacteriophages can heavily impact their infectivity (Douwe et al., 2017; Gonçalo et al., 2018). Furthermore, the half-life of phages and algal viruses under natural conditions ranges from hours to days. In contrast, amoeba-infecting viruses are known for their incredibly stable capsids that can only be opened with very low pH and high temperatures (100 °C) (Schrad et al., 2020). Giant virus virions in permafrost remained infectious for over 30,000 years (Legendre et al., 2015). Because short-term incubations of NiV particles at different temperatures would likely have no effect, we carried out the entire infection experiments at these temperatures.

In summary, NiV could replicate at 20 and 30°C in *Naegleria lovaniensis*, but we did not observe any replication at 40°C. *Naegleria lovaniensis*, devoid of NiV, grew at 40°C. In contrast, *N. lovaniensis* infected with NiV at 40°C did not grow but did not die either. The overall amount of cells stayed the same during the entire experiment, which shows that the presence of the virus suppressed *N. lovaniensis* growth at 40°C.

Given the generally high stability of mimivirus-like virions, combined with a relatively unspecific infection strategy (phagocytosis), and since host fitness of *N. lovaniensis* was impacted by the virus at 40°C, we do not consider the infectivity of the virus but viral replication the limiting factor.

The respective results section now reads: “*Counter to N. clarki, the thermophilic N. lovaniensis grew well at 40 °C in the absence of the virus. In the presence of NiV, N. lovaniensis cell numbers remained notably stable during the entire infection experiment (fold change ~ 1 in Figure 8d, Supplementary Fig. 3). This shows that the virus impaired N. lovaniensis growth at 40°C but did not cause the decline of the amoeba population. Even though NiV suppressed the growth of N. lovaniensis, this did not coincide with an increase in virus abundance. On the contrary, the NiV genome copy numbers fell below the starting numbers (Figure 8b). This indicates that viral particles entered N. lovaniensis trophozoites but could not replicate and were degraded eventually.*”

To further enhance clarity, in the legend of Figure 8 we now note that fold-change values are depicted, and we refer to Supplementary Fig. 3, which includes additional data from the time series experiment: “*NiV replication at 24 hpi, shown as fold change (FC) of NiV genome copy numbers in N. clarki (a) and in N. lovaniensis (b) grown at 20, 30, or 40 °C. Fold change of amoeba cell numbers of N. clarki (c) and N. lovaniensis (d) at 20, 30, and 40 °C with and without NiV at 48 hpi. [...] Additional time points and raw amoeba cell numbers are available as Supplementary Fig. 3.*”

For toxin anti-toxin systems in bacteria, it was shown that temperature affects the functioning of this defense system (Bobonis et al., 2022). Like other enzymes in bacterial cells, defense systems likely are optimized to function in the corresponding cell's growth range. *Naegleria lovaniensis* is a heat-loving amoeba with an optimal growth temperature of around 40 °C, so the antiviral defense systems might function the best at around 40°C. Alternatively, higher temperatures could negatively impact giant virus replication and hinder, for example, the formation of a viral factory or virion assembly. This is now summarized in the discussion section as “*However, our data also suggests that 40°C, the highest temperature tested, impairs viral replication. Whether this is due to a temperature sensitivity of NiV, for instance by obstruction of viral factory formation or virion assembly, or due to host antiviral defense systems, which might function more efficiently at the growth optimum of this heat-loving amoeba, remains unknown.*”

- I. 276: Bodo saltans virus is not italicized, as you refer here to the virus, not the taxon.

Done as suggested.

- I. 292/293: “viral escape mechanism” is used here in the sense of host defense, but could easily be misunderstood in the sense of virions exiting the host cell.

Thank you for pointing out that this sentence was imprecise, we clarified it: “*Couriously, some viral particles appeared to be stuck and degraded in multi-laminar membrane structures (Figures 6a,d), which could represent a potential host defense mechanism.*”

- I. 296-307: very nice interpretation, I completely agree!

Oh, thank you 😊

- I. 326: *N. fowleri* is dangerous to humans because it enters the brain and causes infections there, destroying brain tissue. I fail to see how giant viruses could be used as a treatment, because it would surely be undesirable and also impractical to deliver the virus to the brain. Using the virus in a pre-emptive manner would mean producing large quantities of the virus and treating immense volumes of water on a regular basis with this virus. I do not think that the number of cases justifies such intense and costly measures. Please correct me if I overlooked an important aspect here, but given this brief assessment, I would not emphasize the potential use of NiV as a biocontrol agent, as it may give false hope and seems quite far-fetched. Better concentrate on the interesting virological and evolutionary aspects of your study, which are outstanding by themselves!

We agree that viral control of *N. fowleri* would be challenging, and we share most of the concerns outlined above. We were mostly thinking of applications in smaller water volumes such as private swimming pools but agree that this should not be over-emphasized. We have thus removed the respective statements from the abstract and the introduction, and toned down the respective paragraph in the discussion section, in which this idea is only introduced on a conceptual level. This part of the paragraph now reads:

“[...] N. fowleri abundance in those ecosystems and, thereby, N. fowleri infections and PAM cases are expected to increase. The Naegleriavirus isolate reported here could represent a first step towards a viral biocontrol agent for N. fowleri. Importantly, it allows us to study processes like programmed cell death and antiviral defense in N. fowleri, which will ultimately contribute to a deeper understanding of the biology of this serious human pathogen”.

- Data availability: the genome accession number of NiV should be included in the final version of this paper.

Original sequencing data and the annotated Naegleriavirus genome (accession number OZ003748) are available as BioProject PRJEB72011 at Genbank/ENA/DDBJ.

- Suppl. Fig. 1 legend: replace *Catovirus naegleriensis* (taxon name, has no genome) with Naegleriavirus (trivial name, has a genome)

Done as suggested.

- As there are no supplementary tables 1+2, rename S. tables 3+4 to 1+2.

The Supplementary Information has been reordered and now includes 3 Supplementary Figures and 5 Supplementary Tables.

COMMENTS OF REVIEWER #2

Arthofer et al present a study describing the isolation and characterization of a giant virus that infects hosts in the genus *Naegleria*. This is a well-written study and the figures are clear. The genomic, proteomic, and microscopy-based analyses are comprehensive. It is exciting that a virus infecting the genus *Naegleria* has been found (especially considering it infects the infamous *Naegleria fowleri*). This will be of great interest to those studying giant viruses as well as those who work with protist pathogens, and so it is important that this study is published in a journal with broad readership. The discovery of temperature-dependent virulence is also quite interesting - I have not seen this shown for other systems.

I only have some minor suggestions that may help to improve the manuscript:

Phylum names should be italicized as per ICTV recommendations <https://ictv.global/faqs>

Done as suggested.

Line 104- I am pleased to say that the subfamily Klosneuvirinae has been officially ratified by the ICTV, so “proposed” can be removed.

Thank you, we changed this.

Line 135 - the authors may wish to follow up with an hhpred search - this is more sensitive than interpro and may help confirm that these proteins are mCPs

Following this suggestion, we performed additional remote homology searches using HHpred, which confirmed the annotation of the two minor capsid proteins and revealed similarities to penton and tape measure proteins, respectively. We have included these new results: “*The NiV genome encodes four major capsid domain-containing proteins (CDS 686, 687, 688, and 706) and one major core protein 4b (CDS 720). In addition, two genes were identified as minor capsid proteins using remote homology searches, which suggest they represent a tape measure protein (CDS 824) and a penton protein (CDS 654), respectively.*”

Lines 154-164: the discussion of viral dynamin could include a recent experimental study that shed light on the function of these viral proteins. This study indicated that viral dynamin was involved in remodeling mitochondrial membranes during infection (<https://doi.org/10.1093/molbev/msad134>). Was any obvious changes in the mitochondria apparent during infection here? In addition, the Khalifeh et al study on viral SNAREs, which the authors correctly cite here, postulates that the patchy distribution of these proteins in viruses is indicative of a virus-host antagonistic evolution. In that light, it may be interesting to note whether these proteins are shared with other members of the Klosneuvirinae, or whether they are specific to this genome. Perhaps this could be included in Panel 1D?

Thanks for drawing our attention to this recent study. Indeed, the NiV dynamins are most similar to the recently identified dynamins involved in remodeling of mitochondria. Although we did not observe these effects in our TEM analysis, we have added the reference and modified this section accordingly. We have also added a sentence about the putative origin of the SNARE proteins (which is also discussed in somewhat greater detail in the section about putatively host-derived NiV genes). The revised paragraph reads:
“NiV encodes for two SNARE complex proteins (CDS 76, 229), which enable membrane fusion of transport vesicles and their targets. Furthermore, a vesicle fusing ATPase (CDS 962), dynamin proteins (CDS 14, 15, 373), and additional small GTPases (CDS 237, 396, 417, 494, 531) are present (Supplementary Table 3). The NiV SNARE proteins and the ATPase are most similar to Naegleria proteins. The closest homologs of the small GTPases are eukaryotic, Catovirus or Tupanvirus genes. Dynamin-related proteins of giant viruses have been shown recently to be involved in the remodeling of mitochondrial membranes during infection (Sheikh et al. 2023). Indeed, all three predicted NiV dynamins are similar to other predicted Nuclecytoviricota dynamin-like proteins. Yet, in our TEM analysis we did not observe any striking changes in mitochondria morphology during NiV infection.”

Were any self-splicing introns found? This is a prominent feature of Bodo saltans virus.

We had indeed searched for self-splicing introns using cmscan within Infernal v1.1.4 and suitable Rfam models. While our approach was able to identify the introns in Bodo saltans virus, we didn't detect any in NiV genes. This is now stated in the manuscript (and we added the description of the approach we have used to the methods section): *“Contrary to Bodo saltans virus and other giant viruses, we did not detect any self-splicing introns in NiV.”*

Line 323 - the temperature sensitivity of the virus is quite fascinating. Is higher temperature linked to any alterations in cellular organellar structures? I wonder if high temperatures could interfere with budding of the viral inner membrane from the cellular ER, for example. I find it somewhat harder to believe that host antiviral systems would be upregulated at higher temperatures, though I could be wrong.

Yes, higher temperatures could indeed negatively impact giant virus replication and hinder, for example, the formation of a viral factory or virion assembly. For toxin anti-toxin systems in bacteria, it was shown that there are temperature effects on the functioning of this defense system (Bobonis et al., 2022). Like other enzymes in bacterial cells, defense systems likely are optimized to function in the corresponding cell's growth

range. *Naegleria lovaniensis* is a heat-loving amoeba with an optimal growth temperature of around 40 °C, so the antiviral defense systems might function the best at around 40°C. We have included both ideas in the revised discussion section: *“However, our data also suggests that 40°C, the highest temperature tested, impairs viral replication. Whether this is due to a temperature sensitivity of NiV, for instance by obstruction of viral factory formation or virion assembly, or due to host antiviral defense systems, which might function more efficiently at the growth optimum of this heat-loving amoeba, remains unknown.”*

Line 280: “The virus shares 3.1 % of its genes with its amoeba host” This is an interesting pattern and more details here would be useful. One possibility is that this gene sharing is due to a degraded GEVE present in the host genome. GEVEs are quite common in protist genomes and could explain this pattern (<https://doi.org/10.1038/s41586-020-2924-2>). An alternative explanation would be that the virus acquired these genes from the host, in which case it would be interesting to know of any functional predictions here. Some examples of this are shown in Figure 3, but I wasn't sure if this was the predominant trend or just a few examples. Host genes recently acquired by the virus could be important for specific host-virus interactions and may shed light on the rather narrow host range.

We agree that GEVEs might explain the observed pattern of gene homologs between NiV and host genomes. We added this scenario (and the reference) to the discussion section and provide arguments pointing towards recent gene acquisition as the most parsimonious explanation. The expanded paragraph now reads: *“This pattern of gene sharing could be caused by the integration of viral genomes into those of their hosts, which is widespread among giant viruses infecting algae³⁵. However, endogenization of viral genomes in Naegleria genomes has not been observed to date, and our phylogenetic analysis of the potentially horizontally acquired genes as well as their putative involvement in the manipulation of host cellular pathways point to rather recent acquisitions of host-derived genes into the NiV genome (Figure 3, Supplementary Table 4).”*

In addition, we also added a supplementary table listing the details of all putatively host-derived genes. The putative functional roles of these proteins are summarized in the respective paragraph in the results section: *“The putatively host-derived genes are associated with protein turnover and translation, replication, signal transduction, cell cycle, and apoptosis inhibition (Supplementary Table 4)”*.

For Figure 2, would it not make sense to include a functional category for structural proteins? I was expecting that for the virion proteome part.

We have revised this figure to provide more detail and to facilitate the comparison of genome and proteome data by showing percent of genes/proteins in the genome/proteome for each category. The new figure includes the COG category “Cell wall/membrane/envelope biogenesis”, which includes several structural proteins and is well-represented in the proteome (as opposed to several other categories).

Small point for Figure 3- the authors may wish to increase the line width on the trees, because the thin lines don't show up well on some screens.

Done as suggested.

Figure 7 is somewhat confusing to me given the vast phylogenetic distances between Naegleria (Discoba) and the other protist groups (Amoebozoa) - perhaps the higher-order classifications should be provided here, perhaps as a bar on the right, or put in the text so it is clear that these are completely different phyla/supergroups?

Done as suggested. The updated figure includes boxes to show the affiliations with the Amoebozoa and the Discoba, respectively.

The phylogenetic methods are robust and I am convinced about the placement, so nothing needs to be re-done here, but in the future the authors may wish to know that there are benchmarked workflows available for making multi-locus trees for giant viruses (see https://github.com/faylward/ncldv_markersearch, described in <https://doi.org/10.1371/journal.pbio.3001430>)

Thank you for the recommendation!

Review provided by Frank Aylward

REVIEWERS' COMMENTS

Reviewer #1 (Remarks to the Author):

The authors have addressed all questions in a satisfactory manner.

Reviewer #2 (Remarks to the Author):

My comments have been addressed, and the manuscript is now ready for publication. Congrats to the authors on a very nice study.